# Diverse behaviors in non-uniform chiral and non-chiral swarmalators

Steven Ceron [1,2], Kevin O'Keeffe[3] & Kirstin Petersen [4] ✉

We study the emergent behaviors of a population of swarming coupled oscillators, dubbed swarmalators. Previous work considered the simplest, idealized case: identical swarmalators with global coupling. Here we expand this work by adding more realistic features: local coupling, non-identical natural frequencies, and chirality. This more realistic model generates a variety of new behaviors including lattices of vortices, beating clusters, and interacting phase waves. Similar behaviors are found across natural and artificial micro-scale collective systems, including social slime mold, spermatozoa vortex arrays, and Quincke rollers. Our results indicate a wide range of future use cases, both to aid characterization and understanding of natural swarms, and to design complex interactions in collective systems from soft and active matter to micro-robotics.

Synchronization (self-organization in time) and swarming (self-organization in space) are present in many natural and artificial systems. Synchronization occurs in flashing fireflies[1], firing heart cells[2], spiking neurons[3,4], and chorusing frogs[5]. Swarming, on the other hand, is seen in cell collectives that migrate in response to external signals[6,7], flocks of birds that seamlessly change collective flight direction[8], and schools of fish that coalesce and move together to the collective's advantage[9].

A combination of synchronization and swarming also occurs in diverse contexts, from biological micro-scale collectives[10,11] and chemical micromotors[12,13], to magnetic domain walls[14] and robotic drones[15]. Despite its ubiquity, a theoretical understanding of this interplay between synchronization and swarming is lacking. Synchronization research has, broadly speaking, focused on oscillators which may synchronize in time, but not move around in space[16–19]. Swarming research has done the reverse; it has studied units moving through space[20,21] that synchronize spatially dependent variables like orientation, but not internal phase variables in time. As such, the interplay of synchronization and swarming defines a new kind of collective dynamics about which little is known.

The first steps to explore this domain were taken a few years ago by Igoshin et al.[22], Tanaka et al.[23,24], and Levis et al.[25,26]; they derived models of chemotactic oscillators[22–24] and revolving agents[25,26] and produced diverse behaviors. Later, O'Keeffe et al.[27] proposed a generalized Kuramoto model of 'swarmalators' (short for "swarming oscillators") whose states have been seen in natural[10,11,28] and artificial[13,29–32] micro-scale collectives and is now being extended and implemented on robot swarms[33–38].

This paper continues swarmalator research by introducing a new swarmalator model that facilitates diverse emergent behaviors useful to researchers in fields ranging from bio-inspired swarm robotics to microbotics to biology; our motivation is to comprehensively map out the space of emergent behaviors the model generates. In O'Keeffe et al.[27], for simplicity's sake, the swarmalators were identical, non-chiral, and globally coupled. Here, we relax these idealizations and consider swarmalators in a two-dimensional world which are non-identical (swarmalators have different natural frequencies), chiral (swarmalators have inherent clockwise and counterclockwise circular orbits), and locally coupled (swarmalators can only couple to motion and phase of neighbors within a given radius). Each of these relaxations enhances the model's descriptive power of real systems. (1) Non-identical natural frequencies are representative of systems where agents have differing internal parameters and/or may be noisy. The behaviors of microrobots, for example, vary due to manufacturing tolerances. (2) The chirality in revolving swarmalators enables our model to relate the internal and spatial parameters while mimicking realistic systems where the direction of revolution may be opposite. Following the previous example of microrobots, their orientation (and thus motion patterns) may flip randomly when deposited on a water

[1]Sibley School of Mechanical and Aerospace Engineering, Cornell University, Ithaca, NY 14853, USA. [2]Computer Science and Artificial Intelligence Lab, Massachusetts Institute of Technology, Cambridge, MA 02139, USA. [3]Senseable City Lab, Massachusetts Institute of Technology, Cambridge, MA 02139, USA. [4]Electrical and Computer Engineering, Cornell University, 136 Hoy Road, Ithaca, NY 14853, USA. ✉e-mail: kirstin@cornell.edu

surface. (3) Local coupling ensures that this model is descriptive of distributed systems, where agent sensors or reactive response have a limited range of view. Again, microrobots are a good example, as their physical interactions drop off rapidly with distance. The inclusion of these realistic features enables many new behaviors that to date have not been found in other swarmalator models including interacting phase waves, organized arrays of vortices, concentric phase self-organization, radial oscillation. Many of these behaviors qualitatively resemble those displayed by vortex arrays of sperm[10], the flocking patterns of Quinke rollers[30,31], the various life stages of slime mold[6], spatiotemporal waves of cellular self-organization related to embryology[39–41], and the radial oscillation of phototactic micromotors[12]; the most remarkable states are summarized in Supplementary Fig. 1 and paired with their qualitatively similar real-world counterparts in Supplementary Table 1. Additional states shown in this paper, such as bouncing and revolving clusters, do not to our knowledge have any natural counterparts, but could be useful as inspiration for an applied context. For example, artificial swarmalator systems such as aerial or marine drone collectives could use periodic orbits to maximize surveillance[42] or information-sharing between agents that do not always remain close to each other[43]. Emergent behaviors could also be designed for colloids towards high-precision medicine[44]. We hope this work will inspire new studies on swarmalators; and be useful in the characterization and control of natural and artificial collective systems.

## Results

### The Model

Swarmalators have a spatial position $x_i$ and an internal phase $\theta_i$ which evolve according to Eqs. (1) and (2).

$$\dot{\mathbf{x}}_i = \mathbf{v}_i + \frac{1}{N}\sum_{j\neq i}^{N}\left[\frac{\mathbf{x}_j - \mathbf{x}_i}{|\mathbf{x}_j - \mathbf{x}_i|}\left(A + J\cos\left(\theta_j - \theta_i - Q_{\dot{x}}\right)\right) - B\frac{\mathbf{x}_j - \mathbf{x}_i}{|\mathbf{x}_j - \mathbf{x}_i|^2}\right] \quad (1)$$

$$\dot{\theta}_i = \omega_i + \frac{K}{N}\sum_{j\neq i}^{N}\frac{\sin\left(\theta_j - \theta_i - Q_{\dot{\theta}}\right)}{|\mathbf{x}_j - \mathbf{x}_i|} \quad (2)$$

As per Eq. (1), each oscillator $i$ has an inherent velocity ($\mathbf{v}_i$), a spatial attraction to all other agents defined by a unit vector between agents' positions and a positive coefficient ($A = 1$) which ensures that agents do not dissipate infinitely, a spatial-phase interaction coefficient ($J \in [-1, 1]$) which enables agents with similar phases to move towards each other when $J$ is positive and move away from similarly phased agents when $J$ is negative, and a global repulsive term defined by a power law and a positive coefficient ($B = 1$) which ensures that agents do not aggregate at a single point in space. In Eq. (2), each agent follows the Kuramoto model where there is a natural frequency ($\omega_i$), a phase coupling coefficient ($K$), and an inverse dependence on the distance between agents.

In the original model, all (uncoupled) swarmalators moved in the same direction with $|\mathbf{v}_i| = v_0$, and $v_0$ was set to zero. Here, we modify this by setting $\mathbf{v}_i = c_i\mathbf{n}_i$, so that now the swarmalator model can be generalized to instances where there is a direct mapping between an agent's internal state (phase) and its orientation along a circular orbit.

$$c_i = \omega_i R_i \quad (3)$$

$$\mathbf{n}_i = \begin{bmatrix} \cos\left(\theta_i + \frac{\pi}{2}\right) \\ \sin\left(\theta_i + \frac{\pi}{2}\right) \end{bmatrix} \quad (4)$$

Here, $c_i$ is dependent on an agent's natural frequency ($\omega_i$) and its radius of revolution in real space ($R_i$), and holds a value of −1, 1, or 0

throughout our study. $\mathbf{n}_i$ is a vector pointing in the direction orthogonal to the angle denoted by $\theta_i$, within the global coordinate system. When $|\mathbf{v}_i| = 0$, an agent has no inherent motion and its phase is an internal state; however, when $|\mathbf{v}_i| \neq 0$, an agent follows periodic circular orbits in real space with some inherent radius of revolution and its phase is its angular position about its orbit within the global coordinate system. Our new definition of $\mathbf{v}_i$ enables us to model swarmalator collectives where each agent's phase is either an internal state ($c_i = 0$) or it has a mapping to its position within real space ($c_i = -1$ or $c_i = 1$). $|c_i|$ is the inherent speed of each agent, and for most of this study we only consider collectives where all agents share the same $|c_i|$. When an agent has no inherent motion in real space, $c_i = 0$, $\mathbf{v}_i$ goes away and, throughout the collective, there is no inherent motion (no circular orbiting). $\omega_i$ is never equal to zero in the cases we present; however, past studies have explored the emergent behaviors of swarmalator collectives when $\omega_i = 0$. If there is inherent motion, meaning that $c_i > 0$, then $\omega_i > 0$, this means that the agent's inherent circular orbit within real space is counterclockwise (CCW). If $c_i < 0$, then $\omega_i < 0$, meaning that its inherent circular orbit is clockwise (CW). The nature of our model is similar to recent work by Togashi[45], which maps an agent's internal state/phase to its shape (radius), which affects how it interacts with its neighbors and thus the emergent collective behaviors.

We also include new phase offset terms, $Q_{\dot{x}}$ and $Q_{\dot{\theta}}$, defined in Eqs. (5) and (6), which enable 'frequency coupling'. The motivation here is to increase the attraction between agents with opposing signs for their natural frequency; this enables more realistic emergent behaviors reminiscent of hydrodynamically and mechanically coupled systems[10]. Supplementary Discussion 1 and Supplementary Fig. 2 further discuss the phase offset terms and their effect on the collective's oscillatory behavior.

$$Q_{\dot{x}} = \frac{\pi}{2}\left|\frac{\omega_j}{|\omega_j|} - \frac{\omega_i}{|\omega_i|}\right| \quad (5)$$

$$Q_{\dot{\theta}} = \frac{\pi}{4}\left|\frac{\omega_j}{|\omega_j|} - \frac{\omega_i}{|\omega_i|}\right| \quad (6)$$

Finally, we consider swarmalators with several different cases of natural frequencies ω:

1. Single frequency (F1): $\omega_i = 1$ for all swarmalators.
2. Two frequencies (F2): Exactly half of the swarmalators have $\omega_i = 1$ and the other half have $\omega_i = -1$.
3. Single uniform distribution (F3): All swarmalators have their natural frequency randomly selected from a single uniform distribution, such that $\omega_i \sim U(1, \Omega)$.
4. Double uniform distribution (F4): Exactly half of the swarmalators have their natural frequency randomly selected from one uniform distribution ($\omega_i \sim U(1, \Omega)$) and the second half have their natural frequency selected from another uniform distribution ($\omega_i \sim U(-\Omega, -1)$).

Throughout these definitions and the remainder of the text, $U(X, Y)$ defines a uniform distribution on the interval [X, Y] and $\Omega = 3$ for most of the study; we present additional results in the supplementary material to showcase the collective behaviors when $\Omega$ is higher or lower. The different cases/distributions of natural frequencies are referred to as F1, F2, F3, and F4. The natural frequency distributions F1 and F2 are representative of idealized cases where the collective either shares a single value or is split in half between two equal and opposite values. F3 and F4 represent more disordered cases, common in real-world noisy scenarios. More details on the natural frequency distributions used for this study can be found in Supplementary Discussion 2 and Supplementary Fig. 3.

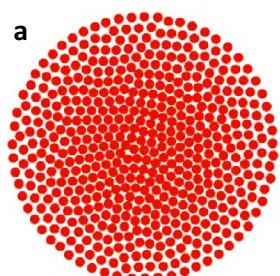

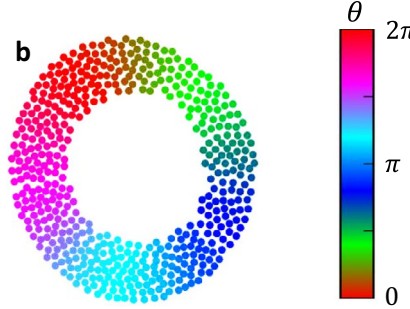

**Fig. 1 | Space-phase order parameter.** Two of the five emergent behaviors in the original swarmalator study. **a** Static sync ($S \approx 0$). **b** Static phase wave ($S \approx 1$). The color bar represents each agent's phase between 0 and $2\pi$; the same color representation is used throughout the work.

To catalog the various collective states of our model, we use several order parameters, some of which were introduced in ref. [27]. The first is,

$$S_{\pm} = \frac{1}{N} \sum_{j=1}^{N} e^{i(\phi_j \pm \theta_j)} \qquad (7)$$

Here, $\phi_j$ is an agent $j$'s angular position with respect to the collective centroid ($\phi_j = (\frac{y_j}{x_j})$), where $x_j$ and $y_j$ are its spatial coordinate positions with respect to the collective centroid, and $\theta_j$ is its phase; this measures a kind of circumferential 'space-phase order' in the system. When there is no correlation between the angular position and phase, as in Fig. 1a, $S_{\pm} = 0$; when there is maximal correlation, as in Fig. 1b, $S_{\pm} = 1$. Throughout our work, we will refer to max ($S_+$, $S_-$) as $S$ and use heat maps of the order parameter in $K - J$ parameter space to draw out the approximate locations of the different emergent states. We also show several plots of the order parameters $S_+$, $S_-$, $Z$, $\gamma$, and $\beta$ (defined in the Methods section), which aid in explaining key differences between the states.

**Non-chiral swarmalators**

We begin our exploration with the simplest case of non-chiral swarmalators with no frequency coupling: $c_i = 0$ for all agents, $Q_{\dot{x}} = 0$, and $Q_{\dot{\theta}} = 0$. We numerically simulated Eqs. (1) and (2) using an Euler method with stepsize $dt = 0.1$ for $T = 1000$ time units at which steady states were achieved. We scanned over the state space $J \in [-1,1]$ and $K \in [-1,2]$. We use the same numerical scheme and parameter regions in $K$ and $J$ space throughout our work.

Numerical simulations revealed a zoo of convergent self-organization behaviors; some stationary, some dynamic. Some of the most interesting are depicted in Fig. 2, the full set are reviewed and shown in Supplementary Discussions 3–4, Supplementary Figs. 4–12, and Supplementary Movie 1. Here, we discuss in detail the properties of each of the most interesting states when the natural frequency distributions are $F1$ and $F2$; however, we strongly recommend viewing Supplementary Movie 1 to have a clear picture of the states in mind before reading this description (we also recommend this for the other sections in the paper). Results regarding the emergent collective behaviors when the system has distributions $F3$ and $F4$ are described in the supplementary material.

The heat map in Fig. 2a shows the general locations of the emergent collective behaviors within $K - J$ parameter space when the natural frequency distribution is $F1$, while Fig. 2b refers to systems where the distribution is $F2$. We note that static async (Fig. 2c) is found through much of negative $K-$ space and phase waves (static, active, and splintered) appear when $K$ is negative and $J$ is positive. This is clear from the bright triangular region in Fig. 2a which indicates there is high circumferential phase organization. Note that $S$ shows very little

difference between the region of static phase waves and splintered phase waves, but dips slightly at low $K$ and high $J$ where the active phase wave emerges. As opposed to previous studies where the static phase wave remained static in both space and phase, the phase wave shown in Fig. 2d has a static annulus formation and a circumferentially traveling phase wave. When there are several natural frequency groups, however, the number of phase waves change, and they become significantly more dynamic. Figure 2e shows two interacting phase waves, each composed of agents with the same natural frequency; the two annulus formations oscillate in place as each of their phase waves travels around the circular border and interacts with the opposing group's phase wave. The two-phase waves shown here are concentric, but it is worth noting that the collective can also settle on an inter-locked ring formation where either of the natural frequency groups can be closer to the collective centroid. Regardless of whether the phase waves are concentric or inter-locked, they remain circumferentially ordered by phase, which enables a high $S$ value along negative $K$ and positive $J$, as shown in Fig. 2b.

At $K \approx -0.1$, $J = 1$, the collective forms a single splintered phase wave when all agents share the same natural frequency, and then display concentric splintered phase waves that rotate about the collective centroid in opposing directions (Fig. 2f). Figure 3 shows that in these cases there is high $S$ while all other calculated order parameters ($Z$, $\gamma$, and $\beta$) remain low. Supplementary Fig. 8 and Supplementary Movie 2 showcase the emergent behaviors when there are two, three, four, or five natural frequency values within the collective. In each of these cases, the collective is evenly split, such that an equal number of agents have each of the natural frequency values, and we find that, with the same $K$ and $J$ values, the number of concentric splintered phase wave formations is correlated to the number of natural frequencies present. When there are more than three natural frequency groups, however, the spacing between the groups is not clear and there is no clear direction of rotation in each layer of the formation.

We also observe the static sync state when there is only one natural frequency (Fig. 2g); however, the collective behavior is significantly more dynamic when the natural frequency distribution is $F2$ because two clusters form and synchronize within themselves, but do not synchronize with the opposing cluster (Fig. 2h). Their inability to synchronize causes the collective to enter a bouncing cluster state where each natural frequency group oscillates between attraction and repulsion with the other group, and the two produce symmetric, periodic oscillations about their mutual centroid. The oscillatory behavior is due to the clusters having equal and opposite natural frequencies; this means one group moves about the phase unit circle in the CW direction while the other moves in the CCW direction. Since both groups have the same absolute natural frequency value, they move about the phase unit circle at the same rate, and share the same phase/are offset by $\pi$ two times per phase cycle. This means the two

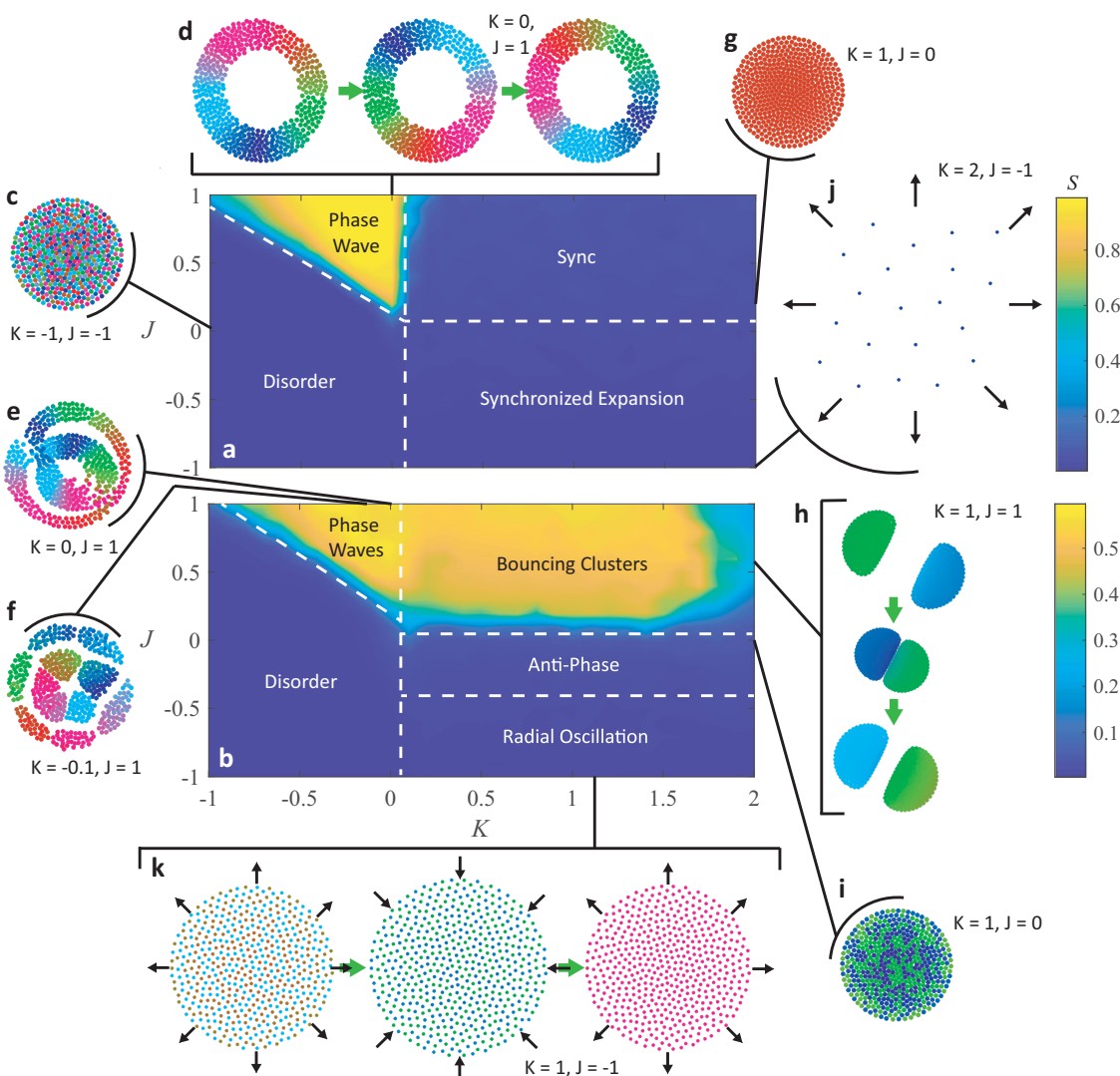

**Fig. 2 | Non-Chiral swarmalators with no natural frequency spread.** Heat maps of $S$ across $K - J$ parameter space are shown for test cases with natural frequency distributions (**a**) $F1$ and (**b**) $F2$. **c** Static async/disorder. **d** Phase wave traveling circumferentially across static agents. **e** Double interacting phase waves. **f** Double interacting splintered phase waves. **g** Static sync. **h** Bouncing clusters. **i** Static anti-phase. **j** Expanding synchronized collective. **k** Periodic radial oscillation.

groups oscillate between having no phase difference and the maximum phase difference and produce periodic attraction and repulsion to each other. Notice in Figs. 2b, 3 that $S$ remains high around low positive $K$, but dips slightly at very high $K$. There is relatively high spatial-phase order for bouncing clusters since the two groups independently synchronize and remain symmetric about an axis running through the collective centroid; Fig. 3b shows that $\beta$ increases as a result of the cluster separation and this allows us to pinpoint the onset of the bouncing cluster behavior when $J$ is positive. At high $K$, the collective eventually transitions from the bouncing cluster state closer to a static sync state, which lowers $S$ because there is no longer circumferential phase organization. The bouncing cluster behavior becomes even more interesting when there are more than two natural frequency groups, a greater number of bouncing clusters form, but their behavior is no longer symmetric about a single axis. Supplementary Fig. 9 and Supplementary Movie 3 show and characterize bouncing cluster collectives when there are more than two natural frequency groups.

At high $K$ and $J = 0$, collectives with the natural frequency distribution $F2$ create circular formations and enter static anti-phase states (Fig. 2i); agents synchronize within their own natural frequency group because of the high phase coupling factor, but remain offset by

phase from the opposing group. Since $J = 0$ ensures that there is no spatial attraction between like-phased agents, there is a fairly uniform distribution of agents from both groups across the formation. The formation is circular because $A = 1$, which enables all agents to attract to each other through a unit vector model, regardless of their phase or distance to each other; the power law model for repulsion ensures that they do not converge to a single point in real space. Once the collective aggregates into a stable circular shape, agents remain fixed and their interaction distance, which still affects their phase coupling behavior, does not change. Some effects of this can be picked out at lower $K$: agents are unable to synchronize and as a result phase waves are seen traveling radially. Agents in the center couple more easily to all other agents, than those close to the boundary; as a result, they are able to lead the phase waves within each natural frequency group. As $K$ gets closer to 2, agents begin to freeze their phase behavior; the two natural frequency groups synchronize within their own group, but maintain a constant phase offset with the opposing group; $K$ is high enough that the two groups phase lock.

Finally, we observe a synchronized state in collectives with a single natural frequency that drives the collective to a circular formation that then proceeds to expand ($K > 0, J < 0$), shown in Fig. 2j. In this case, agents synchronize and then repel from similarly phased agents; since

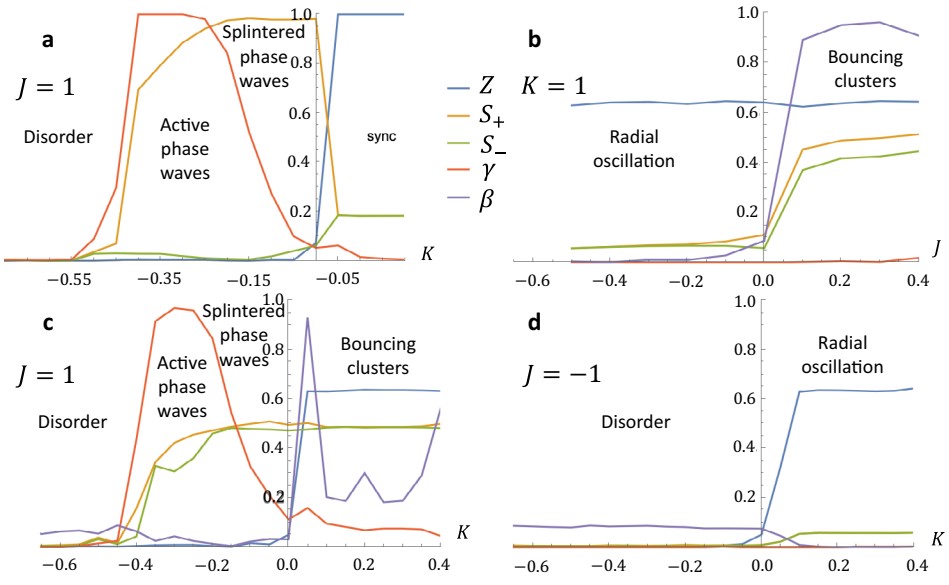

**Fig. 3 | Order parameters for non-chiral swarmalators.** The order parameters $Z$, $S_+$, $S_-$, $\gamma$, and $\beta$ are plotted for non-chiral swarmalators when the natural frequency distribution is $F2$. **a** $J = 1$. **b** $K = 1$. **c** $J = 1$. **d** $J = -1$.

all share the same phase, the repulsion leads to uniform expansion. Supplementary Discussion 4 and Supplementary Fig. 11 address the expansion behavior for various values of $J$ and find that at $J = -1$, the collective expands indefinitely. The same region of $K - J$ parameter space for a natural frequency distribution of $F2$ reveals a radial oscillation state, where the collective evenly distributes agents from both natural frequency groups across a circular formation and proceeds to periodically expand and contract (Fig. 2k). The phase interaction behavior is similar to when there are bouncing clusters; the two groups oscillate between attraction and repulsion, but instead of splitting into multiple clusters, agents coalesce in one so that they are close to agents from the opposing group. As the two groups' phase difference decreases, agents repel each other and the collective expands, but as their phase difference reaches its maximum, attraction occurs and the group contracts. Because agents are close to other agents with a natural frequency of opposite sign, the order parameters $S$ and $\beta$ remain low while $Z$ goes to mid-range, as shown in Fig. 3d.

**Revolving swarmalators**

Revolving swarmalators' motion and phase coupling behavior is defined by Eqs. (1) and (2) when $c_i \neq 0$ for all agents and $Q_{\dot{x}}$, $Q_{\dot{\theta}} = 0$; each agent's natural frequency is defined according to one of the four natural frequency distributions ($F1 - F4$). An overview of the revolving swarmalators' behaviors when there is no frequency coupling is further reviewed in Supplementary Discussions 5–6, shown in Supplementary Movie 6, and characterized in Supplementary Figs. 13–16. Results regarding the emergent collective behaviors when the natural frequency distributions are $F1$ and $F2$ are reviewed in Supplementary Discussion 5.

Revolving collectives composed of agents revolving in the same direction, but with a spread of revolution radii between $\frac{1}{3}$ and 1 are summarized in Fig. 4. Both heat maps demonstrate relatively high circumferential spatial-phase order across the $K - J$ parameter space and especially high $S$ triangular regions spanning positive and negative $K$ and $J$. Disordered phase waves/vortices are also found with a natural frequency spread (Figs. 4c, 5) at low $K$; ordered phases waves lie in the triangular regions. As opposed to most previous phase wave formations where an annulus formed, the frequency spread enables agents to revolve at different inherent radii and form a vortex formation like

the one shown in Fig. 4d, e. However, the vortex in Fig. 4d looks marginally more organized since all agents revolve in the same direction.

The single and multiple synchronized revolving clusters are also found at high $K$ and $J$ (Fig. 4f, g); the main difference is that the agents are more tightly packed when the natural frequency distribution is $F3$ than when it is $F4$. Figure 4f shows a tight cluster revolving and almost reaching phase synchrony, while Fig. 4g shows two more elongated clusters that begin to revolve about each other and never reach phase synchrony. This general revolving cluster behavior persists through negative $J$ as well, the main difference is that the clusters become much sparser (Fig. 4h, i). The $S$ value also dips at high $K$ and low $J$ because the synchronized clusters become so sparse that they occupy wide angular regions of the circular trajectory so the circumferential phase organization decreases.

**Frequency-coupled chiral swarmalators**

Frequency-coupled chiral swarmalators (FCCS) affect each other's motions similar to the way counter-revolving agents might locally affect each other through physical interactions in a real-world setting. In Eqs. (1) and (2), $Q_{\dot{x}}$ and $Q_{\dot{\theta}}$ are defined by Eqs. (5) and (6); therefore, the natural frequency sign difference between two agents plays an important role in how they will affect each other's motion and phase coupling. Because the spatial and temporal interactions are altered by the sign difference between agents' natural frequencies, we only show results for when the natural frequency distributions are $F2$ and $F4$; the emergent behaviors of FCCS when the natural frequency distributions are $F1$ and $F3$ are the same as non-frequency-coupled revolving swarmalators since here all agent have the same sign for their natural frequencies and there is no chirality. The emergent behaviors of FCCS are summarized in Fig. 6 and further discussed and shown in Supplementary Discussion 6, Supplementary Figs. 17–18, and Supplementary Movie 7.

When the collective is split into two equal and opposite natural frequencies, behaviors emerge that appear similar to those from the regular chiral swarmalators; increased repulsion between agents with opposite natural frequency signs, however, makes the collective split into different clusters and minimize the trajectory intersection along

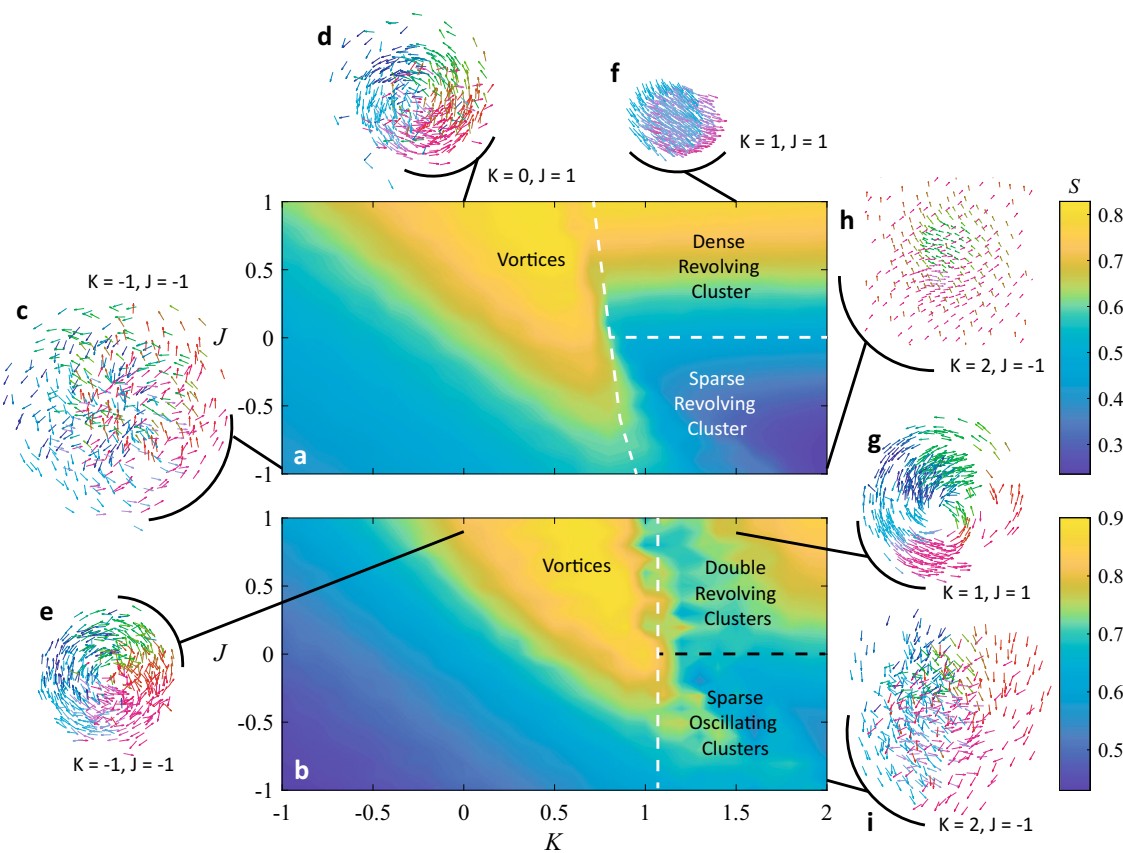

**Fig. 4 | Revolving swarmalators with a natural frequency spread.** Heat maps of $S$ across $K - J$ parameter space shown for test cases with a natural frequency distribution of (**a**) $F3$ and (**b**) $F4$. **c** Disordered vortex. **d**, **e** Vortex. **f**, **g** Revolving clusters. **h**, **i** Sparse revolving clusters.

several regions of the $K - J$ parameter space, shown in Fig. 6a, b. This is not always the case; for example, when the natural frequency distribution is $F2$ and there is low $K$ and $J$, an annulus forms with agents from both natural frequency groups moving close to each other (Fig. 6c). The same region of $K - J$ parameter space when the natural frequency distribution is $F4$ yields high mixing and high disorder (Fig. 6d); the disorder most likely occurs because of the large spread of natural frequencies. The bright rectangular region in the lower half of Fig. 6b indicates a single vortex emerging with high mixing between agents with opposite signs of natural frequency (Fig. 6e).

The first demonstration of high repulsion between counter-revolving agents is shown in Fig. 6f–i. Figure 6f, g shows two interacting phase waves that intersect each other when the natural frequency distribution is $F2$; the non-frequency-coupled swarmalators also exhibit interacting phase waves, but are sometimes difficult to distinguish since they are concentric. The phase wave intersection is caused by the phase shift terms $Q_{\dot{x}}$ and $Q_{\dot{\theta}}$ which enable agents with opposite revolving directions to repel similar phases. At very low $K$, the phase waves become counter-rotating ellipses, which decreases the circumferential phase organization of the whole collective and lowers $S$ (upper left region of Fig. 6a); when $K$ is close to zero, the collective maintains circular double phase waves. When there is a frequency spread, the collective forms two vortices that intersect similarly to the double phase waves; in addition to the vortices, however, there are loose agents around the intersection region that have a high natural frequency and are unable to join the organized formations.

Double revolving clusters are also observed when the natural frequency distribution is $F2$ (Fig. 6k, l); however, FCCS inhibit trajectory intersection for high $K$ and $J$. Rather than follow the same circular trajectory in opposite directions, each cluster follows a fairly circular trajectory, but repels away from the opposing cluster as soon as they begin to intersect. This behavior is much less evident at low $J$ because the clusters become sparser so they can intersect more without being close to other agents.

## Fine-grained analysis

Our analysis so far has been at the global level, the order parameters in Figs. 3 and 5 being single numbers for the entire collective. We supplement this with a finer grained analysis by computing three correlation functions: the velocity auto-correlation function, the position-position correlation function, and the phase-phase correlation function:

$$C_{v,v}(t,\tau) = \left\langle \frac{1}{N} \sum_i \mathbf{v}_i(t) \cdot \mathbf{v}_i(t-\tau) \right\rangle \tag{8}$$

$$C_{x,x}(r) := g(r) := \left\langle \frac{1}{N(N-1)} \sum_{i,j} \delta(r - r_{ij}) \right\rangle \tag{9}$$

$$C_{\theta,\theta}(r) := \left\langle \frac{1}{N(N-1)} \sum_{i,j} \mathbf{n}_i \cdot \mathbf{n}_j \delta(r_{ij} - r) \right\rangle \tag{10}$$

Throughout Eqs. (8)–(10), $\langle \cdot \rangle$ denotes the ensemble average.

Since we have presented a large number of collective states, it is infeasible to calculate the above correlation functions for each state. Instead, we focus on just the sync state and the phase wave/vortex state, which in a sense are the primitives for the other states; for example, the bouncing cluster state is composed of two sync states, and double phase wave is composed of two phase waves.

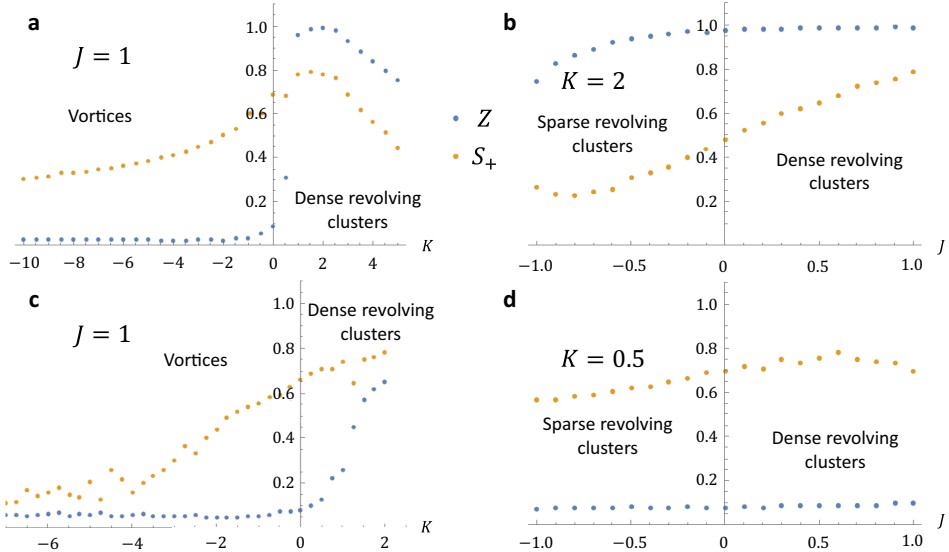

**Fig. 5 | Order parameters for revolving swarmalators.** The order parameters $Z$ and $S_+$ are plotted for chiral swarmalators when the natural frequency distribution is F2 (top row) and F4 (bottom row). **a** $J = 1$, F2 frequency distribution. The dense revolving clusters exist for K > 0 as indicated by $Z, S_+ > 0$. When $K < 0$, the vortex is born in which $Z = 0, S_+ > 0$ (**b**) $K = 2$, F2. Here there is no sharp distinction between the two states, as indicated by both $S_+, Z$ varying smoothly as $J$ increases from −1 to 1. Instead, there is a cross over from sparsity $Z < 1$ to density $Z \approx 1$ (the density in space maps onto a density in phase coherence) (**c**) $J = 1$. F4. (**d**) $K = 0.5$, F4.

The local coupling instance of the chiral swarmalator model is defined by Eqs. (15)–(17): agents only sense other agents within a set radius ($\sigma$). Supplementary Discussions 8–10 and Supplementary Movies 8–12 present a comprehensive analysis of the correlation functions for the sync and phase wave states under various effects, such as dynamical noise $d$ and coupling of finite range $\sigma > 0$. We present the most interesting effects for the sync state here; the phenomena are virtually the same for the phase wave/vortex state (Supplementary Discussions 9 and 10). Supplementary Figs. 20–27 show many of the emergent behaviors for parameters of the local coupling instance, while Supplementary Figs. 28–35 explore how the collective's phase coherence, kinetic energy, and linear momentum fluctuate when there are different natural frequency distributions, a spectrum of $J$ and $K$ values, and a spectrum of $\sigma$ ranging from 0.2 to 2. In Supplementary Figs. 36–39, we also quantify how many clusters appear, the collectives' normalized speed, and the collectives' average normalized angular momentum at different values of $K$ and $J$ for a range $\sigma$ between 0.2 and 2.

Figure 7 shows how the sync state deforms under local coupling. For long range $\sigma = 5$, a single sync cluster is realized in Fig. 7a. The phase-phase correlation function $C_{\theta,\theta}(r) \approx 1$ over this radius, then drops to zero (Fig. 7d). Similarly, the position-position correlation function $g(r)$ has a single peak and drops to zero beyond $r \approx 2$ consistent with a single cluster. For larger noise $d > 0, \Omega > 0$, the sync cluster develops a slight phase gradient reflected in a dip in $C_{\theta,\theta}(r)$ (Supplementary Figs. 54, 55). For intermediary coupling $\sigma = 3$, multiple clusters appear with little inter-cluster phase correlation; $C_{\theta,\theta}(r)$ drops to zero for $r \approx 3$. $g(r)$ develops two new, blurred peaks, which are consistent with a state with multiple clusters (Fig. 7h). As before, adding noise and a frequency spread $d > 0, \Omega > 0$ leads to an intra-cluster phase gradient. Finally for short range coupling $\sigma = 0.5$, a single crystal-like state emerges (Fig. 7c) with length scale $r \approx 10$, as evidenced by the domain of $g(r)$ in Fig. 7i, and local phase coherence, as evidenced by $C_{\theta,\theta}(r)$ having three descending peaks over a length scale $r \approx 2$ (Fig. 7f). This state bifurcates into an incoherent gas-like state when $d > 0$ and $\Omega > 0$.

The same single/multiple cluster and gas-like transition under increasingly local coupling are observed for vortices, both non-chiral (Supplementary Fig. 57) and chiral (Supplementary Fig. 59).

### Real world parallels

The swarmalator research field is still building momentum. While the models demonstrate rich and exciting spatiotemporal patterns[33–38], direct use cases continue to be elusive, earning them a 'toy-model' reputation[46]. To indicate the potential of our model to break this barrier, we use this section to discuss parallels between emergent behaviors in swarmalators and a range of natural and engineered swarms. Figures 8 and 9 show how locally coupled swarmalators qualitatively resemble emergent behaviors in slime mold, vortex arrays of spermatozoa, rotating magnetic colloids, and Quincke rollers, for example. We show additional examples in the Supplementary Material and Supplementary Movies 13–17.

Social slime mold has various life stages that consist of aggregation, collective motion, and dispersal[6,47]. Here, we demonstrate that locally coupled non-chiral swarmalators can qualitatively reproduce these four general life stages of the multi-cellular amoebae. Figure 8a shows the various life stages for *Dictyostelium Discoideum* where thousands of cells begin to aggregate through starvation; as the starvation period continues, prolonged aggregation enables tight clusters that can then combine to form a plasmodium or "slug". This formation enables cells to move together even though there is no central brain within the collective. The slug moves around the environment until it finds nutrients and the collective disperses. Figure 8b shows the swarmalators' version of the general behaviors graphically represented in Fig. 8a; Supplementary Movie 16 shows the transitions between these different behaviors. The non-chiral swarmalators in Fig. 8b change $K$ and $J$ and have the natural frequency distribution F2 and $\sigma = 1.4$. The sparse clusters when $K = 0.05, J = 1$ result from agents that partially synchronize within small groups; when $K = 1, J = 1$, the collectives form tighter clusters. The slug stage is achieved by setting $K = 0, J = 1$; this turns off phase coupling while enabling attraction between agents with a similar phase. A set of phase wave ribbons form instead of the annulus since agents can only sense locally. We have not

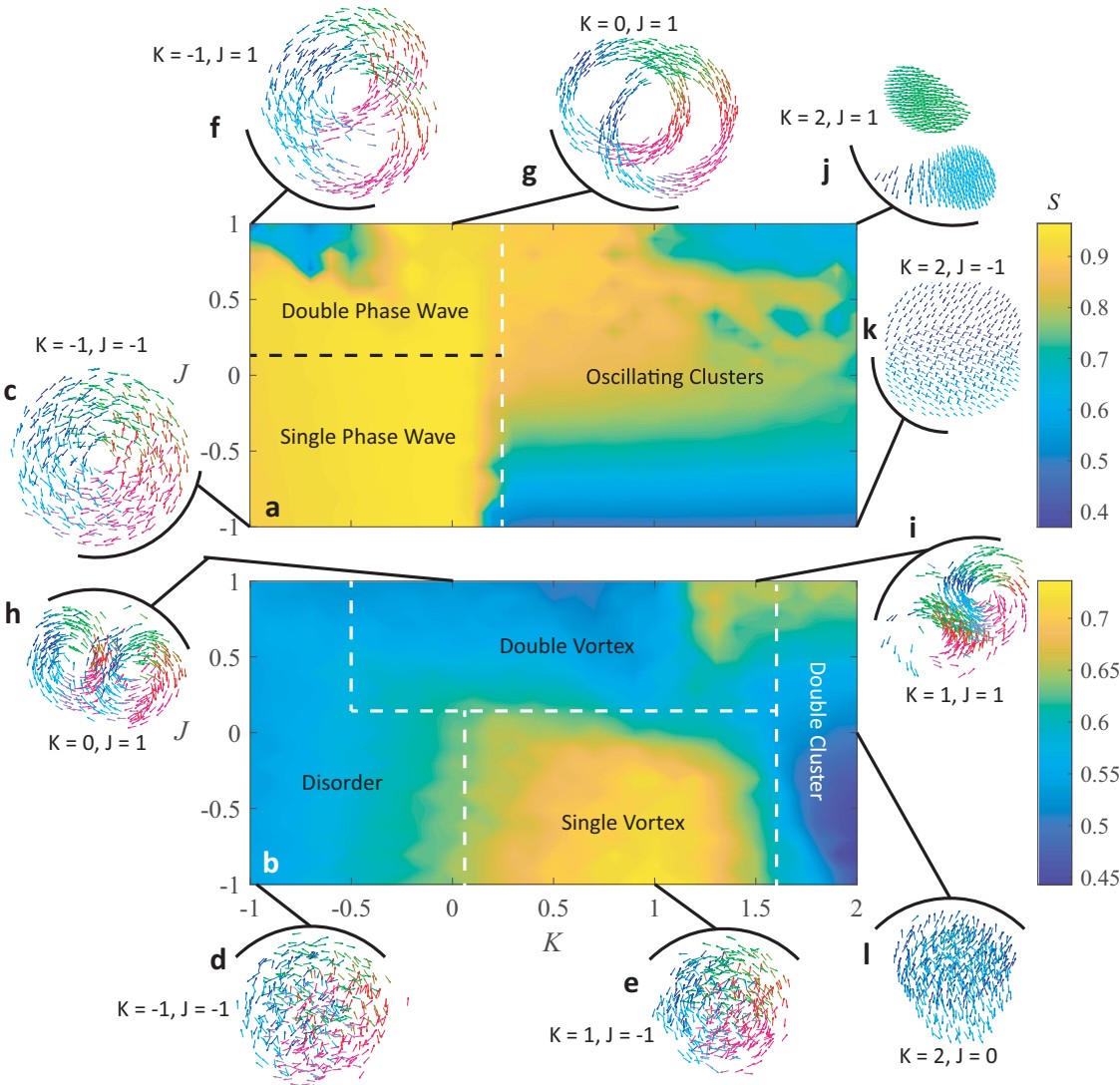

**Fig. 6 | Frequency-coupled chiral swarmalators.** Heat maps of $S$ across $K - J$ parameter space are shown for natural frequency distributions (**a**) $F2$ and (**b**) $F4$. **c** Phase wave. **d, e** Disordered vortices. **f, g** Double phase waves. **h,i** Double vortex. **j** Dense revolving clusters. **k, l** Revolving clusters.

done an exhaustive exploration of state transitions, but we did find that clusters are harder to form when there is high dispersion. Although phase behavior is not the same as what is exhibited in the real social slime mold, we believe this holds great promise for developing complex behaviors with oscillator-like artificial swarms. The non-chiral instance of our model can also be modified as described in Eqs. (16) and (17) to capture the radial phase waves seen in aggregations of embryonic genetics oscillators[41]; this is explored in greater detail in Supplementary Figs. 60–62 and Supplementary Movie 17.

Spinning magnetic microrobots like those shown in Refs. [48–50]. and by Yan et al.[29] produce rotating formations; an image of a rotating microrobot collective[51] is shown in Fig. 8c, and we can create similar vortices (Fig. 8d) with the regular chiral swarmalators. Note that the rotating crystal-like behavior demonstrated in each of these works is not accurately reproduced with the swarmalators because agents are free to move out of their circular trajectory over time, whereas in the real system, the particles in the rotating magnetic colloids are physically enclosed in the crystal-like formation. The microrobots in Gardi et al.[51] are able to move freely, therefore this system is more adequate for realizing swarmalator-like vortices. Indeed, Supplementary Fig. 63 shows that when the collective has an exponential distribution of natural frequencies between zero and one, the speed and angular velocity about the collective centroid are comparable between the

physical experiment and the swarmalator simulations. Following these comparisons, it follows that the general revolving behavior of the swarmalators could be further tuned to study the behavior of other revolving collectives, like those shown by Han et al.[30], Zhang et al.[31], Grzybowski et al. [48,49], and Wang et al.[50].

We also qualitatively compare the formations of the FCCS to Quincke roller collectives[30,31], which exhibit different formations including gas-like, single vortex, multiple vortices, and flocking vortices; these states are graphically depicted in Fig. 8e–g. Figure 8h–j shows sample emergent formations and the corresponding trajectories of swarmalators exhibiting those states when we vary values for $K$, $J$, and the natural frequency distribution. Each of the trajectory snapshots are chosen because they qualitatively compare to formations shown in refs. [30,31]. There is a vast literature on the emergent behaviors of Quincke rollers with many more behaviors that could be worth imitating using the swarmalator models. Finally, Fig. 9 highlights some of the similarities between spermatozoa vortex arrays and the swarmalators when multiple phase waves emerge. Figure 9a shows a graphical representation of the spermatozoa vortex arrays shown by Reidel et al.[10], where the vortices are tightly packed. In their paper, the authors provide a histogram of the sperm head orientation versus position about their circular trajectory; this concept is depicted in Fig. 9b where each dot represents an agent, its color the phase ($\theta_i$) or

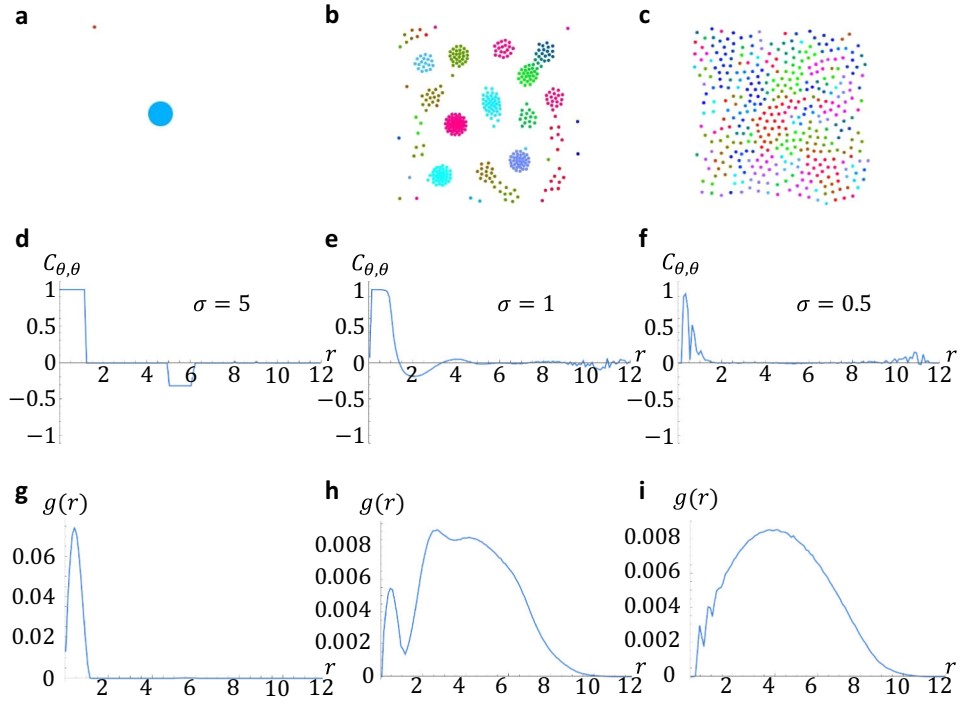

**Fig. 7 | Correlation functions. a–c** Scatter plots of states. **d–f** Phase-phase correlation function for different coupling ranges of $\sigma$. **g, h** $g(r)$ for different coupling ranges of $\sigma$. Simulation parameters: $(J, K, d, \Omega) = (1, 1, 0.05, 0)$, $(dt, T, N) = (0.25, 500, 300)$ with initial position drawn uniformly at random in a box of length $L = 4$ and phases drawn uniformly at random from $[-\pi, \pi]$. **d–i** Each data point is the average of 500 simulations.

position about the circular trajectory, and the orientation is $\phi_i$. We show two similar plots (Fig. 9c, d) and find that the general trend remains similar for swarmalators, at low and high $\sigma$ as shown in Supplementary Fig. 64. This is powerful because we can use this model to replicate some of the emergent collective behaviors of a natural system without considering the specific hydrodynamic coupling mechanisms that occur in the real system. Figure 9e shows similar packing of the ring-like formations; although, as shown by the trajectories in Fig. 9f, the swarmalator vortices are in square grids across all $\sigma$ rather than hexagonal grids like those depicted in Fig. 9a. The FCCS have a strong attraction between agents with differing natural frequency when their phase is offset by $\pi$, so agents from opposing natural frequency groups will have a stronger attraction than agents from the same group. We observe some of the same behavior when the natural frequency distribution is $F4$; the vortex-like formation is not as organized when the natural frequency distribution was $F2$ but vortices with opposing chirality form next to each other because of the attraction between phase offsets of $\pi$.

## Discussion

We explored non-chiral and chiral swarmalators across various natural frequency distributions and found a zoo of emergent behaviors. Non-chiral swarmalators have no inherent motion, yet their temporal behavior drives them towards dynamic behaviors like radial oscillation, synchronized expansion, bouncing clusters, static anti-phase states, concentric phase self-organization, and many variations of phase waves, sync, and partial sync states. Chiral swarmalators couple spatial and temporal behavior by adding an inherent circular motion to each agent; the emergent behaviors follow similar trends to the non-chiral swarmalators in the $K - J$ parameter space, but the circular motion also enables higher levels of spatial-phase order even when there is little or no phase coupling. Some of the most interesting emergent behaviors include single and multiple vortices and phase waves, along with dense and sparse revolving clusters. Throughout the non-chiral and chiral swarmalators, global and local coupling illuminates the effect of distance on the emergent self-organization occurring for different values of $K$ and $J$. The emergent behaviors qualitatively mimic a wide variety of collective systems including slime mold, spermatozoa, magnetic colloids, and Quincke roller collectives.

Past work on swarming systems includes several models similar to the Viscek model[20], which demonstrates that collectives can transition from disordered states to flocks by modifying individual agents' behavior according to their neighbors' motion and orientation. Modified Viscek models defined the agents' motion as inherent revolutions; this led to a breadth of states including revolving flocking patterns[25,26], vortices, and active foams[21]. These models address emergent swarming states in the presence of noise and agent-to-agent coupling; however, if inherent motion is removed, many of the behaviors cease to exist. Our swarmalator model enables us to explore resulting emergent behaviors when the inherent motion is turned on or off. Its strength therefore lies in the fact that it can represent inherently mobile agents where the phase is correlated to orientation, as was the case in the previously mentioned works[20,21,25,26], as well as collectives where there is no inherent motion and the phase is an internal property. To the best of our knowledge, no other active matter model is this concise and offers such a diverse set of emergent collective behaviors. Indeed, many may be finely tuned to closely mimic the behavior of some real world collectives[51]; however, with this new model we can switch between wildly different behaviors by switching just a few global parameters. As shown in the summarizing lists in Supplementary Fig. 1 and Supplementary Table 1, our model covers a wide variety of behaviors that qualitatively resemble many real-world collectives[5,10–12,31,40,47,49–58] and are only partially discovered in other active matter models[20,24–27,36,45].

Given the universality of phase dependence and circular motion, this model may be used to advance studies across fields, from inspiring

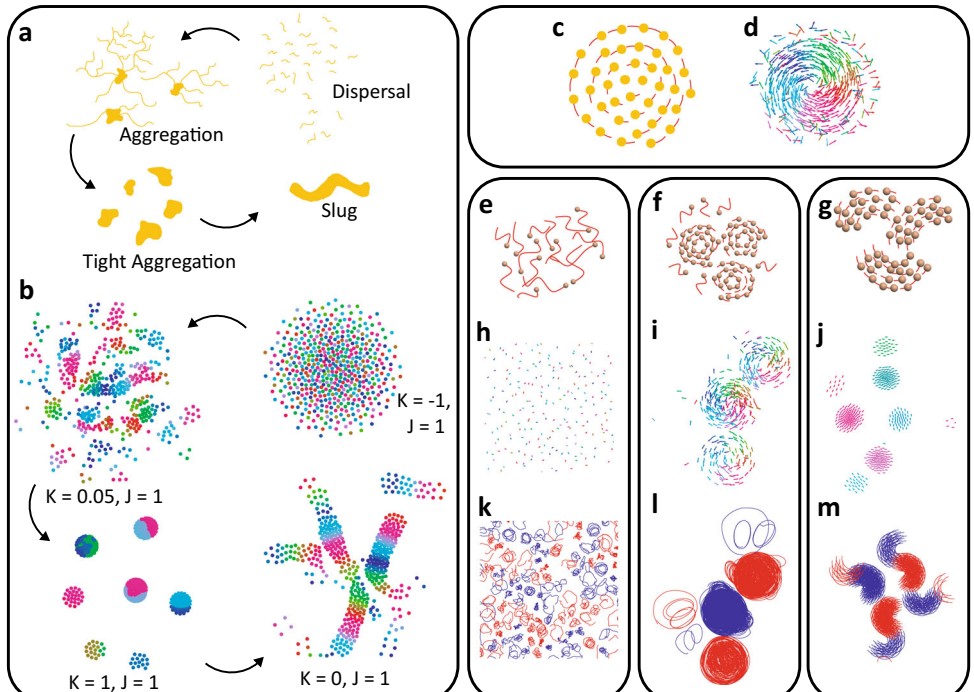

**Fig. 8 | Swarmalators resembling real swarms. a** Graphical representation of slime mold life stages. **b** Stages from (**a**) reproduced by non-chiral swarmalators when $\sigma = 1.4$. **c** Emergent vortex-like formation similar to the one produced in Gardi et al.[51]. **d** Emergent single vortex-like formation qualitatively resembling the one shown in (**c**). **e**–**m** FCCS resembling Quincke rollers. **e**–**g** Graphical representation of Quincke rollers exhibiting three distinct states: gas-like, multiple vortices, and flocking vortices. **h**–**j** Swarmalator behaviors that closely match the behaviors seen in Refs. [30]. The natural frequency distributions are (**h**) F4, (**i**) F4, and (**j**) F2; the parameters here are (**h**) $\sigma = 0.8, K = -1, J = -1$, (**i**) $\sigma = 1.6, K = 0, J = 1$, (**j**) $\sigma = 0.8, K = 1, J = 1$, **k**–**m** Trajectories of swarmalator collectives shown in (**h**–**j**). Blue trajectories correspond to agents with a negative natural frequency and red trajectories to a positive natural frequency.

new theoretical advances in active matter to modeling frameworks in developmental and behavioral biology. These minimalistic coordination schemes may also support macro-scale robot swarms where constituents travel in periodic orbits that exceed their range of communication[43,59], multi-robot systems acting in the human space where more advanced perception and reasoning presents security concerns[60], and microrobot swarms in biomedical applications which are fundamentally limited in both sensing and computation power[51,61,62]. We hope this encourages a line of fundamental studies on the emergent behaviors that result from an interdependence between time-domain and spatial-domain-specific parameters.

## Methods

### General simulation parameters

Numerical studies were run on Matlab using Euler integration, 500 agents, a time step size of $dt = 0.1$, and a final time step of $t_f = 1000$. Simulations for the $S$ order heat maps were run for 10 trials.

### Order parameters

The order parameter $\gamma$ represents the fraction of swarmalators in a collective that have completed at least one cycle of phase and position about the collective centroid after the transient period.

The Kuramoto order parameter $Z$ gives a measure of the overall phase coherence/degree of synchrony. The parameter is defined by the following equation.

$$Z = \frac{1}{N} \sum_{j=1}^{N} e^{i(\theta_j)} \quad (11)$$

The order parameter $\beta$ is the difference in the average position of swarmalators with a positive natural frequency and those with a negative natural frequency, $n_{\omega > 0}$ is the number of swarmalators with a positive natural frequency, $n_{\omega < 0}$ is the number of swarmalators with a negative natural frequency, and $x_{\omega > 0}$ and $x_{\omega > 0}$ are the respective positions of the swarmalators:

$$\beta = \left| \frac{1}{n_{\omega > 0}} \sum_{j=1}^{n_{\omega > 0}} x_{\omega > 0} - \frac{1}{n_{\omega < 0}} \sum_{j=1}^{n_{\omega < 0}} x_{\omega < 0} \right| \quad (12)$$

### Local coupling model

The local coupling model uses step functions to determine whether agents affect each other's motion and phase behavior.

$$\dot{\mathbf{x}}_i = \mathbf{v}_i + \frac{1}{N} \sum_{j \neq i}^{N} \left[ \left( \frac{\mathbf{x}_j - \mathbf{x}_i}{|\mathbf{x}_j - \mathbf{x}_i|} \left( A + J \cos\left(\theta_j - \theta_i - Q_{\dot{x}}\right) \right) \right) - B \frac{\mathbf{x}_j - \mathbf{x}_i}{|\mathbf{x}_j - \mathbf{x}_i|^2} H(\sigma - |\mathbf{x}_j - \mathbf{x}_i|) \right] + \boldsymbol{\xi}_i(t) \quad (13)$$

$$\dot{\theta}_i = \omega_i + \frac{K}{N} \sum_{j \neq i}^{N} \left( \frac{\sin\left(\theta_j - \theta_i - Q_{\dot{\theta}}\right)}{|\mathbf{x}_j - \mathbf{x}_i|} \right) H\left(\sigma - |\mathbf{x}_j - \mathbf{x}_i|\right) + \eta_i(t) \quad (14)$$

$$H\left(\sigma - |\mathbf{x}_j - \mathbf{x}_i|\right) = \begin{cases} 1, & \sigma - |\mathbf{x}_j - \mathbf{x}_i| > 0 \\ 0, & \sigma - |\mathbf{x}_j - \mathbf{x}_i| \leq 0 \end{cases} \quad (15)$$

We can study the effect of white noise on the emergent collective behaviors by adding in the terms $\xi_i$ and $\eta_i$. The white noise terms are $\xi_i$ and $\eta_i$ and are generated from the same distribution. The maximum distance for coupling is defined by the variable $\sigma$.

### Model variation

The non-chiral instance of the model can be slightly modified to demonstrate behavior very similar to embryonic genetic

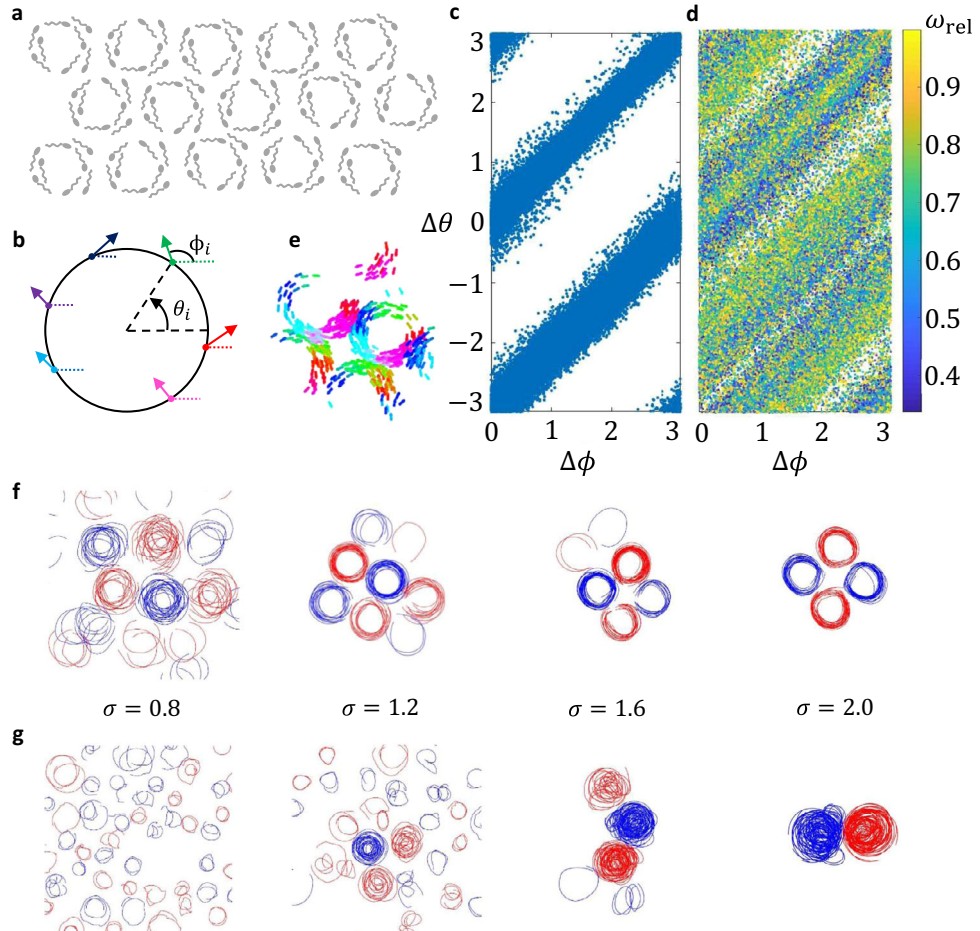

**Fig. 9 | FCCS resembling spermatozoa. a** Graphical representation of sperm vortex arrays. **b** Head orientation vs. position along circular trajectory. **c** Plot of head orientation vs. angular position when the natural frequency distribution is $F2$. **d** Scatter plot of phase (angular position) vs. head orientation color coded by the relative natural frequency; this corresponds to when the natural frequency distribution is $F4$. The result is similar to what is shown in Fig. 2 of Riedel et al.[10]. **e** Snapshot of swarmalators demonstrating this behavior when $K = 0, J = 1, \sigma = 1.2$, and the natural frequency distribution is $F2$. **f** Trajectories of swarmalators self-organizing into arrays of vortices when the natural frequency distribution is $F2$. **g** Trajectories of swarmalators self-organizing into several adjacent vortices when the natural frequency distribution is $F4$.

oscillators.

$$\dot{\mathbf{x}}_i = \mathbf{v}_i + \frac{1}{N}\sum_{j\neq i}^{N}\left(\frac{\mathbf{x}_j - \mathbf{x}_i}{|\mathbf{x}_j - \mathbf{x}_i|^2}\left(A + J\cos\left(\theta_j - \theta_i\right)\right) - B\frac{\mathbf{x}_j - \mathbf{x}_i}{|\mathbf{x}_j - \mathbf{x}_i|^4}\right) + \boldsymbol{\xi}_i(t)$$

(16)

$$\dot{\theta}_i = \omega_i + \frac{K}{N}\sum_{j\neq i}^{N}\sin\left(\theta_j - \theta_i - \alpha\right) + \eta_i(t)$$

(17)

Here, we include a constant phase offset term $\alpha$ and modify the distance relation between swarmalators; $\boldsymbol{v}_i$ is equal to zero for all agents and noise is included in the model.

## Data availability
The paper data is available on Zenodo. Due to the large amount of simulation data, part of the data is available upon request to the corresponding author. The data is available at the following link: https://doi.org/10.5281/zenodo.7554900.

## Code availability
All code is available on Zenodo at the following link: https://doi.org/10.5281/zenodo.7554900.

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

## Acknowledgements

The authors thank Gaurav Gardi for fruitful discussions on possible venues of application for the chiral swarmalators and further analysis that can be done for these swarming systems. We also thank Nils Napp for valuable feedback on how to improve the clarity of this work. S.C. and K.P. thank the National Science Foundation Graduate Research Fellowship, the National Science Foundation grant 2042411, and the Packard Foundation Fellowship for Science and Engineering, and the Aref and Manon Lahham Faculty Fellowship.

## Author contributions

S.C., K.O.K., and K.P. designed the study; S.C. constructed the model, S.C. and K.O.K. performed data processing and analysis; S.C. wrote the manuscript; all authors discussed the results and contributed to the editing of the manuscript; K.P. supervised the research.

## Competing interests

The authors declare no competing interests.
