## [Peer Review File · Nature Communications]

REVIEWER COMMENTS

Reviewer #1 (Remarks to the Author):

Report on manuscript "Diverse Behaviors in Non-Uniform Chiral and Non-Chiral Swarmalators", by S. Ceron et al.

This manuscript reports numerical study of emergent collective behaviour of "swarmalators". Such particles (or agents) exhibit both synchronization and swarming tendencies, and were introduced earlier in ref. [25]. Very rich spatiotemporal organisation of swarmalators was reported as the result of coupling between an internal phase and external spatial dynamics of the agents.

The present study is a significant and non-trivial extension of the earlier research in [25]. The generalizations include: i) non-identical and chiral agents, ii) short-ranged interaction in the real space. The authors show that the inclusion of these realistic features enables the whole zoo of novel time-dependent agent configurations. Examples include vortices, bouncing and rotating aggregates, and various types of phase self-organization. Additionally, the authors were able to identify several configurations which closely resemble patterns observed in some real life systems.

A comment from a more technical point of view, this study represents a very impressive and comprehensive piece of work. Indeed, the employed model is very rich in terms of the number of parameters involved and types of possible behaviours. The authors did a great job by exploring such a large space of parameters and calculating several representative configuration diagrams. Therefore, I am convinced that this study lays a solid foundation for future research activities in areas as diverse as active matter, agents self-organisation, micro-robotics, etc.

Regarding the presentation: The manuscript is mostly clearly written, and the results are well described and interpreted. I have some difficulty understanding in which way the particles are chiral. Is it that they spin in opposite senses, or do the particles follow a clockwise or counterclockwise circular orbits? Please clarify. Additionally, there are several technical questions, listed below, which must be clarified.

In summary, the manuscript contains several interesting new results which not only be of interest for a broad readership, but will also stimulate new research activity in the area of active matter and several related areas. Provided the authors have addressed the questions listed below, I recommend the publication of this manuscript as an article in Nature Communication.

***** Questions to the authors *****

1. Line 76: In the definition of the self-propulsion velocity v_i , is parameter "c" a signed quantity or? Also define "n_i" introduced there.

2. Line 77: why $|w_i|$, and not just w_i , without the module operation. With the absolute value of w_i there is no effect of the sign of w_i on the "c" parameter? Please clarify.

3. Line 78: Is the inherent radius of revolution R_i related to the size of the particles?

4. Line 78: It is not clear why "the swarmalators are now chiral". How does this conclusion follow from the above discussion? Please clarify.

5. Line 79: the statements like "the agent moves in CCW revolutions" is misleading at this stage of the presentation, because it gives an impression that the particles rotate in the real space. However, at this stage in the article, the rotations take place in the inner particle space parametrised by inner variable θ_i . It is only after Eq.(3), that a reader understands that the inner variable θ_i actually defines the particle orientation in the real space. Please clarify.

6. Please explain the motivation behind the terms with w_i in Eq. (3).

7. Line 95: Define function $Uni()$.

8. Line 102: What is x , and y here?

9. Line 115: Does $c=0$ means that the natural frequency $w_i = 0$?

10. Captions of Fig. 2 and Fig.3: Specify Ω .

11. Line 157: Please clarify, is it that each of these more than two groups has a different natural frequency, despite the fact the bimodal frequency distribution was used? Could you discuss through which mechanism these new "non-natural" frequencies emerge?

12. Line 186: Not clear why there is no attraction between like-phases agents, see A-term in eq.(1)? Please clarify.

In any case at $J = 0$ the values of the phases are irrelevant for the attraction between the agents.

13. Line 213: When $c = 1$, it means that there is only one positive natural frequency, according to line 103 in the SI Discussion 2. So, there are particles spinning in just one direction, right? What is chiral here?

14. Line 226: When $R = 0$, then $\omega = \infty$ according to line 102 in SI Discussion 2. Please clarify.

15. Line 226 (and Line 112 in SI Discussion 2): When $R = Uni(0,1)$, does this mean that in $g(\omega) = Uni(1, \Omega)$ $\Omega = \infty$?

16 Line 282: Fig. 4j is not for the frequency distribution specified on the line, but for $Uni(-\Omega, \Omega)$.

17. Lines 395 and 409: What is meant by inherent circular motion, is it phase oscillation (like in Kuramoto model)?

Reviewer #2 (Remarks to the Author):

This paper expands work on swarmilators (swarming coupled oscillators) to include local coupling, non-identical natural frequencies, and chirality. The result is a collection (zoo) of fascinating emergent behaviours demonstrated in simulation and thoroughly outlined in the paper. Parallels are then drawn to emergent behaviours found in nature including slime mould, spermatozoa, magnetic colloids, and Quincke roller collectives.

Overall this is a beautiful and inspiring paper with exciting results for the swarm

community. I expect it will provide a useful tool for swarm engineers, biologists, and micro-engineers alike.

The paper is very well written, helping us navigate the spread of behaviours discovered, and the use of media (video) is instrumental in showcasing the results.

As minor comments, a bit more could be done to make the figures more accessible, highlighting key takehomes and making sure all symbols are defined (e.g. Fig. 6 - include the symbols used in the figure in (c)). Figures are also slightly pixelated.

Furthermore, while videos are gorgeous to watch in conjunction with the paper, they would merit additional labelling to make them stand alone/self-explanatory. E.g. the symbols, or the titles which capture settings being tested could be included in the video. Text is not always well aligned to the images.

Reviewer #3 (Remarks to the Author):

-----General comment-----

The authors study through simulations an active matter model, which is a generalization of the "swarmalator" model introduced in O'Keeffe, K.P. et al. *Oscillators that sync and swarm. Nat Commun* 8, 1504 (2017) (I will call this paper [1]). Their contribution is to introduce chirality, to give every agent a different natural frequency and to introduce frequency coupling. In order to compare simulations with experiments, they change the model interaction from global to local. After the numerical study of their model, they compare the numerical results with experiments.

This paper has some problems and I do not think that in this state it should be published in this journal. I see no clear and quantitative theoretical understanding of the new model as in paper [1] and the study on the simulations consists of a qualitative construction of phase diagrams and little to no quantitative study of all the complex phenomenology that is displayed through videos. No clear quantitative distinction between phases is measured, no rigorous study of vortices has been done and above all the comparison with the experimental systems is qualitative and superficial. I also find some general clarity issues that must be addressed.

I see a possibility of improvement if the authors make a more careful analysis of the local interaction model (which now is only superficially studied), which is the most plausible candidate to be compared with experiments, but it would require a lot of additional work to understand the model and to make a serious comparison with experiments.

-----Crucial issues-----

-General structure and relevance

The swarmalator model was introduced in [1], it is a new interesting and promising model which surely deserves further study. However, in this manuscript there are no theoretical advancements in the understanding of the model, I find that the first part of the paper (up to page 9) describes what the original model of [1] can do if we add complexity in its definition but without really adding any original and profound understanding of the mechanisms that cause all the phenomenology that is observed. I think that this first part, with some improved analysis, could be published in a specific journal about active matter and simulations. It is not of broad interest enough to be published in this journal.

The second part of the manuscript (from the end of page 9 up to the end), where the authors compare the local coupling version of the model with the experiments, is incredibly superficial. The analysis of the local coupling model is poorer than the analysis of the first 9 pages (which is only made on the global coupling model), no phase diagram is presented, no noise is included in the description (hence no correlation functions are measured to see how the local interaction affects correlation) and no serious study of the vortices phenomenon has been made. This second part of the manuscript could be published in this journal only after a serious study about the local model and a quantitative and precise confrontation between observables measured in the local model and in experimental data.

-The definition of frequency distributions is not clear

I am very confused by the methodology the authors use to choose the values of natural frequencies in their work. Above all, I do not understand the need to use uppercase Ω since its meaning is never explained, not even in the Supplementary Information (SI) at page 4, one should stick with lowercase ω . The only clear case is the first one, when all the swarmalators have the same natural frequency $\omega=1$.

In the second case the authors claim that they are using as a distribution for the frequencies a normalized sum of two delta distributions, each delta with the same weight of $1/2$. This means that, in order to fix the frequencies of the swarmalators, they should have chosen with probability $1/2$ either $\omega=1$ or $\omega=-1$ for each particle. For a finite system of N particles this means that you have to flip a coin N times and based on the results of your flip, assign a positive or a negative frequency. What the authors do instead (if I understand correctly) is fixing half of the particles with $\omega=1$ and the other half with $\omega=-1$ which is a quite different thing that does not involve any probability distribution and it is only a particular case (even if the most likely to happen) of the probabilistic scenario coming from the normalized sum of two delta-functions. I recognize that what they do is reasonable but they should fix the notation accordingly, removing the double-delta probability distribution.

In the third case I am even more confused. They claim that they used a uniform distributions for frequencies $\text{Uni}(1, \Omega)$, but once again the value of Ω is not known. Even more confusing is the way they extract values for this distribution. They say in line 9 of SI that they use $\text{Uni}(0,1)$ for the radius R and then $c=1$. After that I assume they compute $\omega=c/R$, as they say in line 3 of SI. With simple probability calculations one finds that such a procedure does not give a uniform distribution for ω , their procedure gives the distribution $g(\omega) = 1/\omega^2$ for ω from 1 to $+\infty$. Hence it is not surprising to find a lot of particles with small frequency and only a small fraction of particles with high frequency. They should either correct the distribution $g(\omega)$ and explain why one should use an asymmetrical power law distribution for frequencies or do again the simulations by extracting directly frequencies from a uniform distribution.

In the fourth case there I find a mix of the two previous problems. Following their practical procedure I understand that they extracted R from $\text{Uni}(0,1)$ and then assigned frequencies for a half of the population $\omega=1/R$ and for the other half $\omega=-1/R$. This means that for the first half of the population the distribution is $g(\omega) = 1/\omega^2$ for ω from 1 to $+\infty$ and for the other half of the population we have $g(\omega) = 1/\omega^2$ for ω from -1 to $-\infty$. This is clearly not $\text{Uni}(-\Omega, \Omega)$ and it should be clarified.

All this confusion should be clarified or a strong discrepancy between what is written with

words and what is written with formulas remains.

-Complete absence of dynamical noise

All the different versions of the model that are simulated in the paper are purely deterministic. If one of the aims of the authors was to give a plausible description of a biological system, they cannot disregard the role of noise, even a simple white Gaussian noise and its role in the system must be studied in order to take into account the stochastic nature of all biological systems. Since the beginning of active matter models, noise has always played an important role, see for example the famous Vicsek model (Vicsek T. et al. Phys. Rev. Lett. 75, 1226 (1995)) and all its derivations (Ginelli, F. The Physics of the Vicsek model. Eur. Phys. J. Spec. Top. 225, 2099–2117 (2016)) or the Kuramoto model (Wüster S. and Bhavna R. Phys. Rev. E 101, 052210 (2020)). One could say that the stochasticity is included in the model via the choice of different frequencies for the particles, but this can be categorized as quenched (or "frozen") disorder and does not contribute dynamically as a random noise added into eq. (1) and (2) would do (see the analysis of [1], eq. (19) and (20)). In my opinion, any model that has the aim to reproduce a biological system must include a dynamical source of stochasticity, but none can be found in the model of the authors.

I find absolutely incredible that in an active matter model, especially for the local interaction case, no noise has been included and hence no correlation function of phases or velocities or positions has been measured. It is a crucial part of the analysis in order to understand the spatial structure of the system in a more profound way, other than simple qualitative descriptions. A few examples of connected correlation functions can be found in Wüster S. and Bhavna R. Phys. Rev. E 101, 052210 (2020) or Ro S. et al. Phys. Rev. Lett. 126, 048003 (2021) or M. C. Marchetti et al. Rev. Mod. Phys. 85, 1143 (2013). I think that without the inclusion of noise and serious analysis of correlation functions (both equal time space correlations and time dependent correlation functions) we cannot even define clearly what is collective behaviour.

-Analysis of simulations

One should be able to understand clearly the difference between all the phases in a system via some quantitative measurement of one or more than one observables. The authors show in the main manuscript only the heat maps about the value of S , but show in the plots a lot of different phases that are not clearly identified by a quantitative measurement of some observable quantity. A few more information about this matter can be found in the SI but it is still not enough to understand precisely how the authors discriminate between all the phases that they have discovered. Even if it is interesting to watch what happens in a particular phase by watching a video of the simulations, it is a bit vague and qualitative. I would like to see a plot in the main manuscript like fig. 6 of [1], where it is extremely clear that the two observables S and γ can describe all the phases of the system and the transition between them. Without a clear study like this, the description of your system remains highly qualitative.

The authors have found that the model, especially in the local interaction case, forms vortices. They must describe this peculiarity of the system in a more systematic way. Some tools to do that are used in the paper (that the authors cite) Han, K. et al. Emergence of self-organized multivortex states in flocks of active rollers. PNAS 117, 9706–9711 (2020) or in this other paper Costanzo, A. & Hemelrijk, C. K. Spontaneous emergence of milling (vortex state) in a Vicsek-like model. J. Phys. D Appl. Phys. 51, 134004 (2018).

-Comparison with experiments

The comparison with experimental data is poor and superficial. No direct comparison with measurable observables has been done. The only thing I can see are images or videos of particles moving but no quantitative analysis to compare results of simulations and experiments has been done. In my opinion this is a big problem and one of the main obstacles that, in my opinion, make this manuscript not fit for publication in this journal. An example of experimental observables that could be measured in simulations, to compare them with real data, could be velocity distributions and angle distributions (As can be found in fig. 4 and 5 of the paper that the authors cite Songa, L. et al. Dictyostelium discoideum chemotaxis: Threshold for directed motion. European Journal of Cell Biology, 85, 981-989 (2006).). As can be seen in the cited paper (Riedel, I. H. et al. A self-organized vortex array of hydrodynamically entrained sperm cells. Science 309, 300-303 (2005)) there are a lot of observables that could be measured to make a comparison with experiments. I am referring to the correlation functions of fig. 2 and also the order parameter χ , that is defined in the text, which measures how much the distribution of the system differs from a binomial distribution. This are only some observables but many others can be found in the experimental results that the authors already cite but do not take into consideration for real quantitative comparisons. A lot of these observables require the introduction of noise in the model, which I think is a crucial ingredient that is missing.

-Wrong claim about experimental comparison

In only one case the authors try to involve a measurable quantity in their analysis. They claims that in (Yan, J., et al. Rotating crystals of magnetic Janus colloids. Soft Matter 11, 147-153 (2015)) they demonstrate " a linear relationship between the crystal's angular velocity and its radius " (line 335 of main manuscript), that is false. As you can see from fig. 3b of the cited paper of Yan et al. there is a linear relation between the inverse of the angular velocity and the square of the radius. Hence fig. 6c of the manuscript demonstrates that the swarmalators model is not coherent at all with the experimental system analyzed by Yan et al. Please clarify about this point.

-Are simulations robust?

All the simulations start from a square homogeneous, densely packed configuration, I would like to know how a change in the initial configuration of your system affects the steady state. Since the authors' aim is to compare the swarmalator model with experimental data, it is crucial to understand how a change in the initial configuration of your system affects the dynamics. For the same reason it would be interesting to see if there are any changes in the system's behaviour with a different number of particles. It is very important that the model still behaves reasonably with slightly more or slightly less particles, since in a real biological system the number of agents may vary.

-----Minor issues-----

- It is obvious for people in the field but, for a broader audience, please state somewhere that your system lives in a 2D space.
- line 76 -> given eq. (3) the normalization in the definition of v_i is useless, n_i already has modulus 1.
- line 80 -> bad practice of referencing to an equation that is written after.
- I think it would be appropriate to cite this interesting recent work in a similar topic, where the internal state of the system is not a phase but the radius of the particle: Togashi, Yuichi. "Modeling of Nanomachine/Micromachine Crowds: Interplay between the Internal State and Surroundings." The Journal of Physical Chemistry B 123.7 (2019): 1481-1490.
- The observables Z_G and Z_{FG} are defined from line 436 to line 446 of the Methods but

they are not used in the main manuscript, maybe it would be more appropriate to define them in the SI directly, since they are only discussed there. Another option could be to include some plots of Z_G and Z_{FG} in the main manuscript, since they could be useful to describe the system's phases.

- There is also a wrong reference to the bibliography in line 333 and 335, it should be 27 instead of 28.

REVIEWER COMMENTS

Reviewer #1 (Remarks to the Author):

Report on manuscript “Diverse Behaviors in Non-Uniform Chiral and Non-Chiral Swarmalators”, by S. Ceron et al.

This manuscript reports numerical study of emergent collective behaviour of “swarmalators”. Such particles (or agents) exhibit both synchronization and swarming tendencies, and were introduced earlier in ref. [25]. Very rich spatiotemporal organisation of swarmalators was reported as the result of coupling between an internal phase and external spatial dynamics of the agents.

The present study is a significant and non-trivial extension of the earlier research in [25]. The generalizations include: i) non-identical and chiral agents, ii) short-ranged interaction in the real space. The authors show that the inclusion of these realistic features enables the whole zoo of novel time-dependent agent configurations. Examples include vortices, bouncing and rotating aggregates, and various types of phase self-organization. Additionally, the authors were able to identify several configurations which closely resemble patterns observed in some real life systems.

A comment from a more technical point of view, this study represents a very impressive and comprehensive piece of work. Indeed, the employed model is very rich in terms of the number of parameters involved and types of possible behaviours. The authors did a great job by exploring such a large space of parameters and calculating several representative configuration diagrams. Therefore, I am convinced that this study lays a solid foundation for future research activities in areas as diverse as active matter, agents self-organisation, micro-robotics, etc.

Regarding the presentation: The manuscript is mostly clearly written, and the results are well described and interpreted. I have some difficulty understanding in which way the particles are chiral. Is it that they spin in opposite senses, or do the particles follow a clockwise or counterclockwise circular orbits? Please clarify. Additionally, there are several technical questions, listed below, which must be clarified.

We thank the reviewer for their positive feedback on our work. The chiral swarmalators follow clockwise and counterclockwise circular orbits. Furthermore, each agent’s orbit radius is dependent on its natural frequency, and in the case of the revolving agents, the natural frequency is the inherent angular velocity about the circular orbit. Agents with a positive natural frequency travel in the counterclockwise direction, and agents with a negative natural frequency travel in the clockwise direction. We have edited the introduction with the following text to clarify that the chiral swarmalators follow clockwise and counterclockwise circular orbits.

“In O’Keefe et al²⁷, for simplicity’s sake, the swarmalators were identical, non-chiral, and globally coupled. Here, we relax these idealizations and consider swarmalators in a two-dimensional world which are non-identical (swarmalators have different natural frequencies), chiral (swarmalators have inherent clockwise and counterclockwise circular orbits), and locally coupled (swarmalators can only couple to motion and phase of neighbors within a given radius).”

In summary, the manuscript contains several interesting new results which not only be of interest for a broad readership, but will also stimulate new research activity in the area of active matter and several related areas. Provided the authors have addressed the questions listed below, I recommend the publication of this manuscript as an article in Nature Communication.

*** Questions to the authors ***

1. Line 76: In the definition of the self-propulsion velocity v_i , is parameter “c” a signed quantity or? Also define “ n_i ” introduced there.

We thank the reviewer for their question on this subject. To make the definition of v_i clearer, we have decided to redefine c and n_i with the following expressions.

$$c_i = \omega_i R_i$$

$$n_i = \left[\cos \left(\theta_i + \frac{\pi}{2} \right) \quad \sin \left(\theta_i + \frac{\pi}{2} \right) \right]$$

We had originally defined c as a constant across the whole collective, that would have a value of $c = 1$ if the collective had inherent circular motions, or $c = 0$ if there was no inherent motion. With the new definition, c_i is now specific to each agent in the collective and will be $c_i = 1$ if the agent follows a circular orbit in the counterclockwise direction, $c_i = -1$ if the agent follows a circular orbit in the clockwise direction, or $c_i = 0$ if the agent holds no inherent circular motion. The self-propulsion velocity is now defined by $v_i = c_i n_i$, where the sign of c_i is dictated by the sign of the natural frequency, and n_i is the vector that directs the motion of the agent along a trajectory orthogonal to its angular position about its circular orbit, which is its phase (θ_i). If $c_i \neq 0$ and positive, then agent i will move in the counterclockwise direction which is the direction dictated by the unit vector n_i . If $c_i \neq 0$ and negative, then the agent will move in the clockwise direction; the negative value of c_i reverses the direction of n_i . In the response to the following question, we expand on our reasoning behind our original definitions for c and n_i and why we have modified those expressions to make the case for chirality clearer.

We have edited the manuscript’s section, ‘The Model’, with the following text to clarify the definition of the self-propulsion velocity, and the significance of c_i and n_i in enabling the model to exhibit chiral and non-chiral characteristics.

“In the original model, all (uncoupled) swarmalators moved in the same direction with $|v_i| = v_0$, and v_0 was set to zero. Here, we modify this by setting $v_i = c_i n_i$, so that now the swarmalator model can be generalized to instances where there is a direct mapping between an agent’s internal state (phase) and its orientation along a circular orbit.

$$c_i = \omega_i R_i \tag{3}$$

$$n_i = \begin{bmatrix} \cos \left(\theta_i + \frac{\pi}{2} \right) \\ \sin \left(\theta_i + \frac{\pi}{2} \right) \end{bmatrix} \tag{4}$$

Here, c_i is dependent on an agent’s natural frequency (ω_i) and its radius of revolution in real space (R_i), and holds a value of -1, 1, or 0 throughout our study. n_i is a vector pointing in the direction orthogonal

to the angle denoted by θ_i , within the global coordinate system. When $v_i = 0$, an agent has no inherent motion and its phase is an internal state; however, when $v_i \neq 0$, an agent follows period circular orbits in real space with some inherent radius of revolution and its phase is its angular position about its orbit within the global coordinate system. Our new definition of v_i enables us to model swarmalator collectives where each agent's phase is either an internal state ($c_i = 0$) or it has a mapping to its position within real space ($c_i = -1$ or $c_i = 1$). $|c_i|$ is the inherent speed of each agent, and for most of this study we only consider collectives where all agents share the same $|c_i|$. When an agent has no inherent motion in real space, $c_i = 0$, v_i goes away and, throughout the collective, there is no inherent motion (no circular orbiting). ω_i is never equal to zero in the cases we present; however, past studies have explored the emergent behaviors of swarmalator collectives when $\omega_i = 0$. If there is inherent motion, meaning that $c_i > 0$, then $\omega_i > 0$, this means that the agent's inherent circular orbit within real space is counterclockwise (CCW). If $c_i < 0$, then $\omega_i < 0$, meaning that its inherent circular orbit is clockwise (CW). The nature of our model is similar to recent work by Togashi⁴⁵, which maps an agent's internal state / phase to its shape (radius), which affects how it interacts with its neighbors and thus the emergent collective behaviors.

We also include new phase offset terms, $Q_{\dot{x}}$ and $Q_{\dot{\theta}}$, defined in Equations (5) and (6), which enable 'frequency coupling'. The motivation here is to increase the attraction between agents with opposing signs for their natural frequency; this enables more realistic emergent behaviors reminiscent of hydrodynamically and mechanically coupled systems¹⁰. Supplementary Discussion 1 and Supplementary Fig. 1 further discuss the phase offset terms and their effect on the collective's oscillatory behavior.

$$Q_{\dot{x}} = \frac{\pi}{2} \left| \frac{\omega_j}{|\omega_j|} - \frac{\omega_i}{|\omega_i|} \right| \quad (5)$$

$$Q_{\dot{\theta}} = \frac{\pi}{4} \left| \frac{\omega_j}{|\omega_j|} - \frac{\omega_i}{|\omega_i|} \right| \quad (6)$$

2. Line 77: why $|\omega_i|$, and not just ω_i , without the module operation. With the absolute value of ω_i there is no effect of the sign of ω_i on the "c" parameter? Please clarify.

We thank the reviewer for pointing out this unclear dependency of c on ω_i . We had originally defined $c = |\omega_i|R_i = 1$; we used $|\omega_i|$ to avoid any dependency on the natural frequency sign so that c could be a constant across the collective with a chosen value of 0 (no inherent motion) or 1 (inherent circular motion), shared by all agents in a collective. Furthermore, we had previously left the direction of travel entirely dependent on n_i , by defining it with the following expression.

$$n_i = \left[\cos \left(\theta_i + \frac{\omega_i \pi}{|\omega_i| 2} \right) \quad \sin \left(\theta_i + \frac{\omega_i \pi}{|\omega_i| 2} \right) \right]$$

In the original definition of n_i , if $\omega_i < 0$, the unit vector pointed in a direction orthogonal to an agent's angular position about its circular orbit or phase (θ_i) in the clockwise direction. If $\omega_i > 0$, then the agent moves in the counterclockwise direction along a circular orbit.

We have decided to clean up the definitions of c and n_i by changing them to the following two expressions and have updated the manuscript text accordingly.

$$c_i = \omega_i R_i$$

$$n_i = \left[\cos \cos \left(\theta_i + \frac{\pi}{2} \right) \quad \sin \sin \left(\theta_i + \frac{\pi}{2} \right) \right]$$

The new definition of c is specific to each agent so it is now c_i , and it can have a positive or negative value. The new definition of n_i omits any dependency on ω_i so that it is now always a vector that is offset by positive $\frac{\pi}{2}$ radians to an agent's phase, θ_i . An agent's direction of travel in the clockwise or counterclockwise direction is now left to c_i , which will hold a value of $-1, 0, 1$. We have clarified these new definitions by editing the manuscript with the text shown in response to question 1.

3. Line 78: Is the inherent radius of revolution R_i related to the size of the particles?

We thank the reviewer for bringing this to our attention. Each agent is a point particle; therefore, its size is zero. An agent's inherent radius of revolution (R_i) is defined by the following expression, where c_i holds a value of 1 if the agent has an inherent circular orbit in the counterclockwise direction, or a value of -1 if it has an inherent motion in the clockwise direction.

$$R_i = \frac{c_i}{\omega_i}$$

Following this expression, a larger natural frequency (ω_i), or inherent angular velocity, means that the inherent radius of revolution will be smaller; a smaller ω_i means that R_i will be larger.

4. Line 78: It is not clear why “the swarmalators are now chiral”. How does this conclusion follow from the above discussion? Please clarify.

We appreciate the feedback for the needed changes in this section of the paper. Following the new definitions for c_i and n_i addressed in questions 1 and 2, it is now clearer that the swarmalators are chiral if c_i is non-zero throughout the collective, with a value of 1 for some agents and a value of -1 for other agents. We have modified the manuscript text, as shown in the response to question 1, to explain the new definitions of c_i and n_i and to make it more clear how there can be chiral swarmalators.

5. Line 79: the statements like “the agent moves in CCW revolutions” is misleading at this stage of the presentation, because it gives an impression that the particles rotate in the real space. However, at this stage in the article, the rotations take place in the inner particle space parametrised by inner variable θ_i . It is only after Eq.(3), that a reader understands that the inner variable θ_i actually defines the particle orientation in the real space. Please clarify.

We appreciate the reviewer's concern with the clarity of the article at this stage; we have redefined c and n_i in the previous questions and have edited that portion of the manuscript, as shown in the text in response to question 1, so that the new definitions for c_i and n_i appear before we explain instances where the swarmalators follow circular orbits.

6. Please explain the motivation behind the terms with w_i in Eq. (3).

In our original submission, we defined n_i in Eq. (3) so that it dictated the direction of motion of an agent i . In the old definition of n_i , $\frac{\omega_i}{|\omega_i|}$ held a value of -1 or 1 ; therefore, an agent with a negative ω_i would have an inherent motion in the clockwise direction, and an agent with a positive ω_i would have an inherent motion in the counterclockwise direction. We have redefined n_i , so it is no longer dependent on $\frac{\omega_i}{|\omega_i|}$.

In our responses to questions 1 and 2, we have redefined c and \mathbf{n}_i so that they now have the following definitions.

$$c_i = \omega_i R_i$$

$$\mathbf{n}_i = \left[\cos\left(\theta_i + \frac{\pi}{2}\right) \sin\left(\theta_i + \frac{\pi}{2}\right) \right]$$

The modified text in response to question 1 addresses the new definitions of c_i and \mathbf{n}_i and the role that ω_i plays within them.

7. Line 95: Define function Uni().

We realize that Uni() was better represented by U(X,Y), which signifies a uniform distribution between the values X and Y. We have defined U() in the text with the following lines:

“Finally, we consider swarmalators with several different cases of natural frequencies ω :

1. Single frequency (F1): $\omega_i = 1$ for all swarmalators
2. Two frequencies (F2): Exactly half of the swarmalators have $\omega_i = 1$ and the other half have $\omega_i = -1$.
3. Single uniform distribution (F3): All swarmalators have their natural frequency randomly selected from a single uniform distribution, such that $\omega_i \sim U(1, \Omega)$.
4. Double uniform distribution (F4): Exactly half of the swarmalators have their natural frequency randomly selected from one uniform distribution ($\omega_i \sim U(1, \Omega)$) and the second half have their natural frequency selected from another uniform distribution ($\omega_i \sim U(-\Omega, -1)$).

Throughout these definitions and the remainder of the text, $U(X, Y)$ defines a uniform distribution on the interval $[X, Y]$ and $\Omega = 3$ for most of the study; we present additional results in the supplementary material to showcase the collective behaviors when Ω is higher or lower.”

8. Line 102: What is x , and y here?

In line 102, φ_j is the spatial angular position of agent j about the collective’s centroid, and it is defined in terms of x and y . We have renamed x and y as x_j and y_j , respectively, to specify that they are the spatial coordinates of agent j with respect to the collective’s centroid. We thank the reviewer for helping us clarify this for the readers. The following text has been added to the main text to clarify the meaning of x_j and y_j .

“Here, φ_j is an agent j ’s angular position with respect to the collective centroid ($\varphi_j = \left(\frac{y_j}{x_j}\right)$), where x_j and y_j are its spatial coordinate positions with respect to the collective centroid, and θ_j is its phase; this measures a kind of circumferential ‘space-phase order’ in the system.”

9. Line 115: Does $c=0$ means that the natural frequency $\omega_i = 0$?

We thank the reviewer for pointing out this unclear relationship between c and ω_i . Following our new definition for c_i in the responses to the previous questions ($c_i = \omega_i R_i$), when $c_i = 0$, ω_i is not necessarily equal to zero since c_i is also dependent on R_i ; however, if $\omega_i = 0$, then $c_i = 0$. Because R_i represents an agent’s radius of revolution, it must either be positive or zero, therefore, c_i and ω_i will always share the same sign if $R_i > 0$.

There are three possible cases for the value of c_i : (1) $c_i = 0$, (2) $c_i < 0$, and (3) $c_i > 0$.

Case (1): When $c_i = 0$, it can be that $\omega_i = 0$, or $R_i = 0$, or both are equal to zero. If $R_i = 0$, then agents do not follow circular orbits and as a result their phase (θ_i) is simply a representation of their internal state.

Case (2): When $c_i < 0$ then $\omega_i < 0$

Case (3): When $c_i > 0$ then $\omega_i > 0$

10. Captions of Fig. 2 and Fig.3: Specify Omega.

We thank the reviewer for their recommendation. In Fig. 2, the top portion of the figure is devoted to swarmalator collectives where all agents share the same natural frequency, and the bottom portion of the figure refers to collectives where half of the collective has a natural frequency equal to $-1 \frac{rad}{s}$ and the other half has a natural frequency equal to $1 \frac{rad}{s}$. This figure refers to collectives where $c_i = 0$ throughout the whole collective such that there are no inherent circular motions.

Fig. 3 in the original manuscript is now Fig. 4 in the revised submission and refers to collectives where $c_i \neq 0$ throughout the whole collective, which means there are inherent circular motions. The top portion is devoted to when $\omega_i \sim U(1,3)$ and the bottom portion is devoted to when half the collective has $\omega_i \sim U(1,3)$ and the other half has $\omega_i \sim U(-3, -1)$.

Each of the captions has been updated so that they reference the appropriate natural frequency distribution; four natural frequency distributions were tested throughout the paper: $F1$, $F2$, $F3$, and $F4$.

The caption for Fig. 2 reads:

“Fig. 2. Non-Chiral swarmalators with no natural frequency spread. Heat maps of S across $K - J$ parameter space are shown for test cases with (a) Natural frequency distribution $F1$ and (b) natural frequency distribution $F2$. (c) Static async / disorder. (d) Phase wave traveling circumferentially across static agents. (e) Double interacting phase waves. (f) Double interacting splintered phase waves. (g) Static sync. (h) Bouncing clusters. (i) Static anti-phase. (j) Expanding synchronized collective. (k) Periodic radial oscillation.”

The caption for Fig. 4 reads:

“Fig. 4. Revolving swarmalators with a natural frequency spread. Heat maps of S across $K - J$ parameter space are shown for test cases with a natural frequency distribution of (a) $F3$ and (b) $F4$. (c) Disorder. (d-e) Vortex. (f-g) Revolving clusters. (h-i) Sparse revolving clusters.”

11. Line 157: Please clarify, is it that each of these more than two groups has a different natural frequency, despite the fact the bimodal frequency distribution was used? Could you discuss through which mechanism these new “non-natural” frequencies emerge?

We appreciate the reviewer bringing this to our attention. For the sake of clarity, in Fig. 2 of the main text, we only showcase the emergent behaviors when the whole collective has $\omega_i = 1$ and a second case where half of the collective has $\omega_i = -1$ and the other half has $\omega_i = 1$.

Here, we are referring to emergent collective behaviors presented only in the supplementary material, where there may be more than two natural frequencies in the collective. Supplementary Fig. 7 showcases the splintered phase waves that result when there are two, three, four, or five natural frequency values in

the collective; $K = -0.1$ and $J = 1$ are maintained for each formation shown. The supplementary figure caption lists the natural frequency values in each case. Note that when there are two natural frequencies, the collective is split in equal halves between the two natural frequency values, when there are three natural frequencies, the collective is split in equal thirds, when there are four it is split in equal fourths, and for five it is split in equal fifths.

Supplementary Fig. 8 also shows the emergent behaviors when $K = 1$ and $J = 1$, which enables multiple bouncing clusters. This supplementary figure caption also details the specific natural frequency values present in the collective.

We would like to make very interested readers aware that there are many more emergent behaviors waiting to be explored when the collective has more than two natural frequencies, but we are reserving the analysis on these behaviors for a future study of its own.

We have edited the manuscript at that location to clarify that the results for more than two natural frequency groups is reserved for the supplementary materials:

“At $K \approx -0.1$, $J = 1$, the collective forms a single splintered phase wave when all agents share the same natural frequency, and then display concentric splintered phase waves that rotate about the collective centroid in opposing directions (Fig. 2f). Fig. 3 shows that in these cases there is high S while all other calculated order parameters (Z , γ , and β) remain low. Supplementary Fig. 7 and Supplementary Movie 2 showcase the emergent behaviors when there are two, three, four, or five natural frequency values within the collective. In each of these cases, the collective is evenly split, such that an equal number of agents have each of the natural frequency values, and we find that, with the same K and J values, the number of concentric splintered phase wave formations is correlated to the number of natural frequencies present. When there are more than three natural frequency groups, however, the spacing between the groups is not clear and there is no clear direction of rotation in each layer of the formation.”

12. Line 186: Not clear why there is no attraction between like-phases agents, see A-term in eq.(1)? Please clarify.

We thank the reviewer for bringing this issue with clarity to our attention. Line 186 of the original submission refers to the static anti-phase state, where $K = 1$ and $J = 0$ and the collective forms a single cluster with one half of the collective out of phase with the other half of the collective. When $K = 1$, the equation of phase behavior (Eq. 2) ensures that agents with the same natural frequency will couple to each other and eventually share similar phases, but as stated by the reviewer, the values of the phases are irrelevant for the attraction between agents when $J = 0$. The equation of motion has two main components, one component has a unit vector for attraction between all agents and a power law model for repulsion between all agents. The unit vector component additionally has two sub-components which are controlled by A and $J\cos(\theta_j - \theta_i)$. The second sub-component is partially dependent on the phase values of the agents, but the first subcomponent (A) is just dependent on the scalar value, which is always kept at a value of 1. $A = 1$ and $J = 0$ ensures that all agents will be attracted to each other with the same ‘force’ regardless of phase. Because of the repulsion from the power law model portion of the equation of motion, the collective settles at a circular formation with relatively uniform mixing between the two natural frequency groups. We have clarified this confusion by editing the manuscript with the following lines.

“At high K and $J = 0$, collectives with the natural frequency distribution $F2$ create circular formations and enter static anti-phase states (Fig. 2i); agents synchronize within their own natural frequency group

because of the high phase coupling factor, but remain offset by phase from the opposing group. Since $J = 0$ ensures that there is no spatial attraction between like-phased agents, there is a fairly uniform distribution of agents from both groups across the formation. The formation is circular because $A = 1$, which enables all agents to attract to each other through a unit vector model, regardless of their phase or distance to each other; the power law model for repulsion ensures that they do not converge to a single point in real space.”

13. Line 213: When $c = 1$, it means that there is only one positive natural frequency, according to line 103 in the SI Discussion 2. So, there are particles spinning in just one direction, right? What is chiral here?

We appreciate the reviewer pointing this out. We realize that it is incorrect to label the collective as chiral when all agents revolve in the same direction; we have fixed this definition throughout the paper and have made sure to clarify the distinction between a chiral and a revolving collective. We now appropriately refer to collectives in which all agents are revolving in the same direction as “revolving collectives” and collectives in which some agents are revolving clockwise and others count clockwise are referred to as “chiral”.

14. Line 226: When $R = 0$, then $\omega = \infty$ according to line 102 in SI Discussion 2. Please clarify.

We thank the reviewer for pointing out this problem. Following the third reviewer’s concerns on the nature of the natural frequency distributions throughout our work, we have rerun the study with natural frequency distributions between 1 and 3. This means that the maximum radius of revolution will be 1 and the minimum radius of revolution will be $\frac{1}{3}$ when there is a natural frequency spread. We find that this makes a difference in some of the states found throughout our work; however, the new natural frequency distribution is now a uniform distribution and more representative of real swarms.

15. Line 226 (and Line 112 in SI Discussion 2): When $R = \text{Uni}(0,1)$, does this mean that in $g(\omega) = \text{Uni}(1, \Omega) \setminus \Omega = \infty$?

As stated in our response to question 14, we have altered our distribution of natural frequencies which means that each agent’s radius of revolution now has a uniform distribution between $\frac{1}{3}$ and 1.

16 Line 282: Fig. 4j is not for the frequency distribution specified on the line, but for $\text{Uni}(-\Omega, \Omega)$.

We thank the reviewer for catching this mistake. We have renamed the different natural frequency distributions as $F1$, $F2$, $F3$, and $F4$ and have made sure to refer to the correct distributions throughout the paper.

17. Lines 395 and 409: What is meant by inherent circular motion, is it phase oscillation (like in Kuramoto model)?

We thank the reviewer for bringing this confusion to our attention. Inherent circular motion refers to the circular orbit in real space that each agent takes when c_i is non-zero. When $c_i \neq 0$, the agent has a mapping between its orientation along its circular orbit and its phase; its phase (θ_i) is its angular position about its circular orbit. When $c = 0$, the radius of revolution is zero, which means that θ_i is simply an

internal phase state with no mapping to its position in real space. We have clarified this confusion by editing the manuscript with the text in response to question 1.

Final remarks:

We appreciate all the reviewer's comments and questions; we believe they have elevated the clarity of our article and have made us present a cleaner representation of chiral and non-chiral swarmalators. The new definitions of c_i and n_i have made us consider the possibility of collectives where some agents have a circular motion and other agents that do not have a circular motion. In the original submission, the whole collective was either composed of agents that all had inherent circular motions or agents that had no inherent circular motion, but never mixed. Now that c_i is specific to each agent, we can test the emergent collective behaviors when some of the agents have inherent circular motion and others have no inherent circular motion. We believe the results we have shown with collectives where all agents have equal $|c_i|$ is sufficient for the purposes of our study; however, we believe that many more fascinating collective behaviors will result in future studies where $|c_i|$ is not the same throughout the collective.

Reviewer #2 (Remarks to the Author):

This paper expands work on swarmalators (swarming coupled oscillators) to include local coupling, non-identical natural frequencies, and chirality. The result is a collection (zoo) of fascinating emergent behaviours demonstrated in simulation and thoroughly outlined in the paper. Parallels are then drawn to emergent behaviours found in nature including slime mould, spermatozoa, magnetic colloids, and Quincke roller collectives.

Overall this is a beautiful and inspiring paper with exciting results for the swarm community. I expect it will provide a useful tool for swarm engineers, biologists, and micro-engineers alike.

The paper is very well written, helping us navigate the spread of behaviours discovered, and the use of media (video) is instrumental in showcasing the results.

We appreciate the reviewer's positive assessment of our work and for recognizing the great potential applications of this work in a variety of fields.

As minor comments, a bit more could be done to make the figures more accessible, highlighting key takehomes and making sure all symbols are defined (e.g. Fig. 6 - include the symbols used in the figure in (c)). Figures are also slightly pixelated.

We appreciate the reviewer's comment and have updated the figures throughout the paper to make sure they have better quality and include any relevant information needed to understand the figure by itself.

Furthermore, while videos are gorgeous to watch in conjunction with the paper, they would merit additional labelling to make them stand alone/self-explanatory. E.g. the symbols, or the titles which capture settings being tested could be included in the video. Text is not always well aligned to the images.

We thank the reviewer for their positive assessment of our videos. We have made changes to the labeling on the videos and have updated the relevant videos with the new results arising from our change in natural frequency distribution.

Reviewer #3 (Remarks to the Author):

-----General comment-----

The authors study through simulations an active matter model, which is a generalization of the "swarmalator" model introduced in O’Keeffe, K.P. et al. Oscillators that sync and swarm. Nat Commun 8, 1504 (2017) (I will call this paper [1]). Their contribution is to introduce chirality, to give every agent a different natural frequency and to introduce frequency coupling. In order to compare simulations with experiments, they change the model interaction from global to local. After the numerical study of their model, they compare the numerical results with experiments.

This paper has some problems and I do not think that in this state it should be published in this journal. I see no clear and quantitative theoretical understanding of the new model as in paper [1] and the study on the simulations consists of a qualitative construction of phase diagrams and little to no quantitative study of all the complex phenomenology that is displayed through videos. No clear quantitative distinction between phases is measured, no rigorous study of vortices has been done and above all the comparison with the experimental systems is qualitative and superficial. I also find some general clarity issues that must be addressed.

I see a possibility of improvement if the authors make a more careful analysis of the local interaction model (which now is only superficially studied), which is the most plausible candidate to be compared with experiments, but it would require a lot of additional work to understand the model and to make a serious comparison with experiments.

-----Crucial issues-----

-General structure and relevance

The swarmalator model was introduced in [1], it is a new interesting and promising model which surely deserves further study. However, in this manuscript there are no theoretical advancements in the understanding of the model, I find that the first part of the paper (up to page 9) describes what the original model of [1] can do if we add complexity in its definition but without really adding any original and profound understanding of the mechanisms that cause all the phenomenology that is observed. I think that this first part, with some improved analysis, could be published in a specific journal about active matter and simulations. It is not of broad interest enough to be published in this journal.

We appreciate the reviewer’s view on this. However, we believe that while mostly descriptive, our detailed mapping of emergent behaviors across the large parameter space may hold value both to shape more targeted theoretical advancements and to guide experimental biologists, physicists, and roboticists to make use of the swarmalator model, even before we fully understand the underlying fundamentals. Furthermore, we believe that given the numerous new states, and the fact that there is a very broad audience (roboticists, mathematicians, physicists, biologists, etc.) that would find the new states interesting and valuable, it seems reasonable to leave the first part of the paper in the main text. Reviewer 1 and 2 for example believe the discovery of these state will “lay a solid foundation for future research”:

“The authors did a great job by exploring such a large space of parameters and calculating several representative configuration diagrams. Therefore, I am convinced that this study lays a solid foundation

for future research activities in areas as diverse as active matter, agents self-organisation, micro-robotics, etc.”

“Overall this is a beautiful and inspiring paper with exciting results for the swarm community. I expect it will provide a useful tool for swarm engineers, biologists, and micro-engineers alike. “

That said, we agree that in the original submission, not much was included to understand the differences between the different emergent states; however, in our revised submission, we have included new order parameter plots and a deeper analysis of some of the behaviors which allows us to better understand the mechanisms for the emergence of these states which makes this first part of the manuscript even more compelling.

The second part of the manuscript (from the end of page 9 up to the end), where the authors compare the local coupling version of the model with the experiments, is incredibly superficial. The analysis of the local coupling model is poorer than the analysis of the first 9 pages (which is only made on the global coupling model), no phase diagram is presented, no noise is included in the description (hence no correlation functions are measured to see how the local interaction affects correlation) and no serious study of the vortices phenomenon has been made. This second part of the manuscript could be published in this journal only after a serious study about the local model and a quantitative and precise confrontation between observables measured in the local model and in experimental data.

We appreciate this comment and the following comments related to our analysis and comparison to experiments. To address these issues, we have added a significant amount of material to the main text and supplementary material to address many of these issues. Specifically, we have added 1) deeper analysis through correlation functions, 2) characterized the vortex-like behavior, 3) included noise in our local interaction cases, 4) provided a comparison with a recent microrobotics paper's collective, and 5) further characterized the local agent interaction cases through several parameters. We believe our new additions have significantly elevated the clarity and quality of the paper.

-The definition of frequency distributions is not clear

I am very confused by the methodology the authors use to choose the values of natural frequencies in their work. Above all, I do not understand the need to use uppercase Ω since its meaning is never explained, not even in the Supplementary Information (SI) at page 4, one should stick with lowercase ω . The only clear case is the first one, when all the swarmalators have the same natural frequency $\omega=1$.

We appreciate the reviewer's concern about our use of Ω and agree that in the original submission there was no need for the symbol and its meaning was confusing. We have now made use of it in our additional analysis of the states in the main text and supplementary material. We especially make use of it throughout Supplementary Discussions 8, 9 and 10; throughout these sections and the corresponding supplementary figures, we vary the value of Ω to change the size of the uniform distribution across which the natural frequencies of the swarmalators are chosen.

In the second case the authors claim that they are using as a distribution for the frequencies a normalized sum of two delta distributions, each delta with the same weight of 1/2. This means that, in order to fix the frequencies of the swarmalators, they should have chosen with probability 1/2 either $\omega=1$ or $\omega=-1$ for each particle. For a finite system of N particles this means that you have to flip a coin N

times and based on the results of your flip, assign a positive or a negative frequency. What the authors do instead (if I understand correctly) is fixing half of the particles with $\omega=1$ and the other half with $\omega=-1$ which is a quite different thing that does not involve any probability distribution and it is only a particular case (even if the most likely to happen) of the probabilistic scenario coming from the normalized sum of two delta-functions. I recognize that what they do is reasonable but they should fix the notation accordingly, removing the double-delta probability distribution.

We thank the reviewer for this comment and agree that the notation was slightly misleading. We are indeed giving one half of the collective a natural frequency value of 1 and the other half a natural frequency value of -1 in the cases corresponding to this comment. We have fixed this issue by stating that half the collective has $\omega_i = 1$ and the other half have $\omega_i = -1$.

Furthermore, we have listed the different cases for the collective's natural frequencies as distributions *F1*, *F2*, *F3*, and *F4*. We define each of these cases in "The Model" section with the following text:

"Finally, we consider swarmalators with several different cases of natural frequencies ω :

5. Single frequency (*F1*): $\omega_i = 1$ for all swarmalators
6. Two frequencies (*F2*): Exactly half of the swarmalators have $\omega_i = 1$ and the other half have $\omega_i = -1$.
7. Single uniform distribution (*F3*): All swarmalators have their natural frequency randomly selected from a single uniform distribution, such that $\omega_i \sim U(1, \Omega)$.
8. Double uniform distribution (*F4*): Exactly half of the swarmalators have their natural frequency randomly selected from one uniform distribution ($\omega_i \sim U(1, \Omega)$) and the second half have their natural frequency selected from another uniform distribution ($\omega_i \sim U(-\Omega, -1)$).

Throughout these definitions and the remainder of the text, $U(X, Y)$ defines a uniform distribution between X and Y and $\Omega = 3$ for most of the study; we present additional results in the supplementary material to showcase the collective behaviors when Ω is higher or lower. The different cases / distributions of natural frequencies are referred to as *F1*, *F2*, *F3*, and *F4*. More details on the natural frequency distributions used for this study can be found in Supplementary Discussion 2 and Supplementary Fig. 2."

In the third case I am even more confused. They claim that they used a uniform distributions for frequencies $\text{Uni}(1, \Omega)$, but once again the value of Ω is not known. Even more confusing is the way they extract values for this distribution. They say in line 9 of SI that they use $\text{Uni}(0, 1)$ for the radius R and then $c=1$. After that I assume they compute $\omega=c/R$, as they say in line 3 of SI. With simple probability calculations one finds that such a procedure does not give a uniform distribution for ω , their procedure gives the distribution $g(\omega) = 1/\omega^2$ for ω from 1 to +infinity. Hence it is not surprising to find a lot of particles with small frequency and only a small fraction of particles with high frequency. They should either correct the distribution $g(\omega)$ and explain why one should use an asymmetrical power law distribution for frequencies or do again the simulations by extracting directly frequencies from a uniform distribution.

We appreciate the reviewer's comment; indeed, while this method did allow us to make sure all agents had maximum revolution radius of 1, the resulting natural frequencies were not in a uniform distribution and several had very high values. Although this produced many interesting behaviors, we decided that it was unrealistic and that it would make it more difficult for diverse readers to make useful comparisons to behaviors in real systems.

Taking this into consideration, we decided it was better to assign natural frequencies between two values 1 and 3 ($\omega_i \sim U(1, 3)$). This ensured that the natural frequencies in the collective had a uniform distribution while also ensuring the inherent revolution radius of each swarmalator was somewhere between 1/3 and 1. We have rerun all relevant simulations with this new method, and refer to this case throughout the paper as natural frequency distribution *F3*, as stated in the excerpt from the main text in the response to the preceding comment. After running these simulations again, we found that this changed a few behaviors and have updated the corresponding plots, figures, and text in the main text and supplementary material.

In the fourth case there I find a mix of the two previous problems. Following their practical procedure I understand that they extracted R from $\text{Uni}(0,1)$ and then assigned frequencies for a half of the population $\omega = 1/R$ and for the other half $\omega = -1/R$. This means that for the first half of the population the distribution is $g(\omega) = 1/\omega^2$ for ω from 1 to $+\infty$ and for the other half of the population we have $g(\omega) = 1/\omega^2$ for ω from -1 to $-\infty$. This is clearly not $\text{Uni}(-\Omega, \Omega)$ and it should be clarified.

We appreciate the reviewer pointing this out. We agree that our method for generating the natural frequencies and the corresponding notation needed to be fixed. Our method for generating the natural frequencies and our reasoning behind it can be found in our response to the two preceding questions. In this case, some emergent behaviors did change with the new distribution and we have updated all corresponding plots, figures, and text in the main text and supplementary material.

All this confusion should be clarified or a strong discrepancy between what is written with words and what is written with formulas remains.

-Complete absence of dynamical noise

All the different versions of the model that are simulated in the paper are purely deterministic. If one of the aims of the authors was to give a plausible description of a biological system, they cannot disregard the role of noise, even a simple white Gaussian noise and its role in the system must be studied in order to take into account the stochastic nature of all biological systems. Since the beginning of active matter models, noise has always played an important role, see for example the famous Vicsek model (Vicsek T. et al. Phys. Rev. Lett. 75, 1226 (1995)) and all its derivations (Ginelli, F. The Physics of the Vicsek model. Eur. Phys. J. Spec. Top. 225, 2099–2117 (2016)) or the Kuramoto model (Wüster S. and Bhavna R. Phys. Rev. E 101, 052210 (2020)). One could say that the stochasticity is included in the model via the choice of different frequencies for the particles, but this can be categorized as quenched (or "frozen") disorder and does not contribute dynamically as a random noise added into eq. (1) and (2) would do (see the analysis of [1], eq. (19) and (20)). In my opinion, any model that has the aim to reproduce a biological system must include a dynamical source of stochasticity, but none can be found in the model of the authors.

We agree that adding dynamical noise would be an improvement, so we have done so in several places in our revised manuscript. However, we note that in the synchronization phenomena at least, quenched disorder and active disorder are in fact often dynamically equivalent. See:

[1] Tönjes, Ralf, and Arkady Pikovsky. "Low-dimensional description for ensembles of identical phase oscillators subject to Cauchy noise." *Physical Review E* 102.5 (2020): 052315.

[2] Tanaka, Takuma. "Low-dimensional dynamics of phase oscillators driven by Cauchy noise." *Physical Review E* 102.4 (2020): 042220.

I find absolutely incredible that in an active matter model, especially for the local interaction case, no noise has been included and hence no correlation function of phases or velocities or positions has been measured. It is a crucial part of the analysis in order to understand the spatial structure of the system in a more profound way, other than simple qualitative descriptions. A few examples of connected correlation functions can be found in Wüster S. and Bhavna R. *Phys. Rev. E* 101, 052210 (2020) or Ro S. et al. *Phys. Rev. Lett.* 126, 048003 (2021) or M. C. Marchetti et al. *Rev. Mod. Phys.* 85, 1143 (2013). I think that without the inclusion of noise and serious analysis of correlation functions (both equal time space correlations and time dependent correlation functions) we cannot even define clearly what is collective behavior.

We agree analyzing correlation functions would benefit our work. So we have followed the referee's recommendations and included a lengthy analysis of various correlation functions under various effects (detailed below).

We note however that since we have presented a 'zoo of collective states' in our paper, it is infeasible to compute a set of correlation functions for *every* state / bifurcation of states. Instead, we have taken the following targeted approach which captured a large swathe of the phenomenology we discovered; the manuscript now stands at 20 pages, with nine figures, while the supplementary material is 81 pages with 16 movies – any larger analysis is beyond the scope of this paper.

Analysis of correlation functions

We experimented with various correlation functions, and found that the following were the most informative for our study:

1. Velocity auto-correlation functions (time dependent) ($C_{v,v}(t, \tau)$):

$$C_{v,v}(t; \tau) := \left\langle \frac{1}{N} \sum_i \hat{v}_i(t) \hat{v}_i(t + \tau) \right\rangle$$

2. Position-position correlation function (time independent) ($C_{x,x}(r)$):

$$C_{x,x}(r) := g(r) := \left\langle \frac{1}{N(N-1)} \sum_{i,j} \delta(\mathbf{x} - \mathbf{x}_{ij}) \right\rangle$$

3. Phase-phase correlation function (time independent) ($C_{\theta,\theta}$):

$$C_{\theta,\theta}(r) := \left\langle \frac{1}{N(N-1)} \sum_{i,j} \hat{n}_i \cdot \hat{n}_j \delta(\mathbf{x}_{ij} - \mathbf{x}) \right\rangle$$

Where $n_i = \left(\cos\left(\theta_i + \frac{\pi}{2}\right), \sin\left(\theta_i + \frac{\pi}{2}\right) \right)$, \hat{v} is a unit vector in the direction of the spatial velocity, and $\langle \cdot \rangle$ denotes the ensemble average. In fact, the position-position and phase-phase correlations were the most informative, but we include velocity auto-correlation, as was suggested by the referee, to include a time dependent correlation function.

Since we have discovered a zoo of collective states, it was infeasible to compute $C_{v,v}$, $g(r)$, and $C_{\theta,\theta}$ for every collective state, so we focused on the two main states:

- a. sync
- b. active phase wave / vortex

which are in some sense the ‘base’ states from which the other states are composed; for instance, the bouncing cluster state can be seen as two interacting ‘sync’ states, and the “double phase wave states” can be seen as two interacting ‘active phase wave / vortex states’. Furthermore, we computed the dependence of these correlation functions on four effects / parameters

- (i) Active disorder: $d > 0$
- (ii) Quenched disorder: $\Omega > 0$
- (iii) Local coupling: $\sigma > 0$
- (iv) Chirality: $c > 0$ when there are positive and negative natural frequencies within the collective.

In summary, we computed correlation functions 1,2,3 for states a), b) under effects (i) (ii) (iii) (iv). We report this comprehensive analysis in the main text in the ‘Analysis’ section, and in Supplementary Discussions 8-10.

-Analysis of simulations

One should be able to understand clearly the difference between all the phases in a system via some quantitative measurement of one or more than one observables. The authors show in the main manuscript only the heat maps about the value of S , but show in the plots a lot of different phases that are not clearly identified by a quantitative measurement of some observable quantity. A few more information about this matter can be found in the SI but it is still not enough to understand precisely how the authors discriminate between all the phases that they have discovered. Even if it is interesting to watch what happens in a particular phase by watching a video of the simulations, it is a bit vague and qualitative. I would like to see a plot in the main manuscript like fig. 6 of [1], where it is extremely clear that the two observables S and γ can describe all the phases of the system and the transition between them. Without a clear study like this, the description of your system remains highly qualitative.

We appreciate the reviewer pointing this out to us; we agree that defining more order parameters beyond just the heat maps of S could help the reader more easily understand the difference between each of the states. Therefore, we have updated the main text to include two order parameter plots (Figs. 3 and 5) that include the order parameters Z , S_+ , S_- , γ , and β . Z is the degree of synchrony, S_+ and S_- are the circumferential space-phase order, γ is the fraction of swarms that have executed at least one full cycle of phase and position about the collective centroid, and β is the difference in the average position of swarms with a positive natural frequency and with a negative natural frequency. These plots allow us to better explain the quantitative differences between many of the states.

The authors have found that the model, especially in the local interaction case, forms vortices. They must describe this peculiarity of the system in a more systematic way. Some tools to do that are used in the paper (that the authors cite) Han, K. et al. Emergence of self-organized multivortex states in flocks of active rollers. PNAS 117, 9706–9711 (2020) or in this other paper Costanzo, A. & Hemelrijk, C. K.

Spontaneous emergence of milling (vortex state) in a Vicsek-like model. J. Phys. D Appl. Phys. 51, 134004 (2018).

We agree with the reviewer that it is important to describe the emergent vortices more systematically in the local interaction cases. We appreciate the papers that were listed and have used two parameters calculated in “Spontaneous emergence of milling (vortex state) in a Vicsek-like model.” J. Phys. D Appl. Phys. 51, 134004 (2018) to describe the behavior of the vortices across increasing interaction radii in Supplementary Figs. 35 – 38. Throughout these figures we show how the frequency distribution, chirality, and radius of interaction affects the number of clusters that form, the normalized velocity, and the normalized angular momentum for different values of K , when $J = 1$.

-Comparison with experiments

The comparison with experimental data is poor and superficial. No direct comparison with measurable observables has been done. The only thing I can see are images or videos of particles moving but no quantitative analysis to compare results of simulations and experiments has been done. In my opinion this is a big problem and one of the main obstacles that, in my opinion, make this manuscript not fit for publication in this journal. An example of experimental observables that could be measured in simulations, to compare them with real data, could be velocity distributions and angle distributions (As can be found in fig. 4 and 5 of the paper that the authors cite Songa, L. et al. Dictyostelium discoideum chemotaxis: Threshold for directed motion. European Journal of Cell Biology, 85, 981-989 (2006).). As can be seen in the cited paper (Riedel, I. H. et al. A self-organized vortex array of hydrodynamically entrained sperm cells. Science 309, 300–303 (2005)) there are a lot of observables that could be measured to make a comparison with experiments. I am referring to the correlation functions of fig. 2 and also the order parameter χ , that is defined in the text, which measures how much the distribution of the system differs from a binomial distribution. This are only some observables but many others can be found in the experimental results that the authors already cite but do not take into consideration for real quantitative comparisons. A lot of these observables require the introduction of noise in the model, which I think is a crucial ingredient that is missing.

We appreciate this comment and agree that the comparisons with experiments could be improved. In the main text (Fig. 8d) we have provided a scatter plot similar to the histogram shown in Fig. 3d of Riedel, I. H. et al. A self-organized vortex array of hydrodynamically entrained sperm cells. Science 309, 300–303 (2005). Furthermore, using experimental data from the microrobot collectives in Ref. 50, we show in Supplementary Fig. 59 that the swarmalator collectives can match trends in 1) speed vs normalized distance from the collective centroid and 2) angular velocity about collective centroid vs. distance from the collective centroid, if the correct natural frequency distribution is chosen for the collective.

We note however that the swarmalator model here presented is intended as a *minimal model*, whose purpose is to capture the *qualitative* features of real world swarmalators. We see the model here in the tradition of say the Kuramoto model, which imitates the essential features of many self-synchronizing systems (spiking neurons, flashing fireflies, neutrino oscillations), yet captures the quantitative features of far fewer systems (only Josephson Junctions spring come to mind). To give a concrete example, the kuramoto model predicts a $r \sim (K - K_c)^{-\frac{1}{2}}$ scaling for the order parameter, which in fact is highly non-generic and springs from the purely sinusoidal coupling. If higher order harmonics are included – the more general case, for example used to approximate Hodgekin Huxley neurons in neuroscience – then the scaling becomes $r \sim (K - K_c)^{-1}$ – the Kuramoto model does **not** quantitatively predict the general case.

We have reworded the main text in several places to make it clear that our model matches the mentioned experimental systems only / mainly qualitatively.

-Wrong claim about experimental comparison

In only one case the authors try to involve a measurable quantity in their analysis. They claim that in (Yan, J., et al. Rotating crystals of magnetic Janus colloids. *Soft Matter* 11, 147–153 (2015)) they demonstrate " a linear relationship between the crystal's angular velocity and its radius " (line 335 of main manuscript), that is false. As you can see from fig. 3b of the cited paper of Yan et al. there is a linear relation between the inverse of the angular velocity and the square of the radius. Hence fig. 6c of the manuscript demonstrates that the swarmalators model is not coherent at all with the experimental system analyzed by Yan et al. Please clarify about this point.

We really appreciate the reviewer pointing out this mistake to us. We apologize for the mistaken linear relationship and have decided to omit that quantitative comparison and instead state in the main text that the swarmalator collective produces a vortex-like state that qualitatively resembles the experimental behavior. We also state in the main text that there are other systems, like the microrobot collective used in Ref. 50, that may better serve as real-world examples of swarmalators since these are freely able to move along a plane, whereas the magnetic colloid is tightly packed in a crystal-like configuration.

Furthermore, since we have now characterized many of the emergent states and demonstrated in the supplementary material that the swarmalators can produce similar trends to the revolving behavior shown by the microrobot collectives in Ref. 50 (Gardi, G.*, Ceron, S.*, Wang W., Petersen, K., Sitti, M. Microrobot collectives with reconfigurable morphologies, behaviors, and functions. *Nature Communications* 13, 2239 (2022).), we have decided to not move forward in attempting to reproduce the trends in Ref. 29 (Yan, J., Bae, S. C. & Granick, S. Rotating crystals of magnetic Janus colloids. *Soft Matter* 11, 147–153 (2015). We do state in the main text, however, that the swarmalator model could potentially be used to study many collective behaviors if the correct natural frequency distribution is chosen, as was the case when we compared the swarmalators to the microrobot collective in Supplementary Fig. 59.

-Are simulations robust?

All the simulations start from a square homogeneous, densely packed configuration, I would like to know how a change in the initial configuration of your system affects the steady state. Since the authors' aim is to compare the swarmalator model with experimental data, it is crucial to understand how a change in the initial configuration of your system affects the dynamics. For the same reason it would be interesting to see if there are any changes in the system's behaviour with a different number of particles. It is very important that the model still behaves reasonably with slightly more or slightly less particles, since in a real biological system the number of agents may vary.

We appreciate this comment and agree with the reviewer that it is important to make sure that the system behaves predictably independent of the initial configuration. Therefore, throughout our system analysis of the behavior when d and Ω are changed, we change the number of swarmalators and the box side length within which the swarmalators are placed at the start of each simulation.

-----Minor issues-----

- It is obvious for people in the field but, for a broader audience, please state somewhere that your system lives in a 2D space.

We appreciate this comment and have made it clear that the system lives in a 2D space by making the addition to the following text in the introduction section of the main text.

“In O’Keeffe et al²⁷, for simplicity’s sake, the swarmalators were identical, non-chiral, and globally coupled. Here, we relax these idealizations and consider swarmalators in a two-dimensional world which are non-identical (swarmalators have different natural frequencies), chiral (swarmalators have inherent clockwise and counterclockwise circular orbits), and locally coupled (swarmalators can only couple to motion and phase of neighbors within a given radius).”

- line 76 -> given eq. (3) the normalization in the definition of v_i is useless, n_i already has modulus 1.

We thank the reviewer for this comment and have revised our definition of n_i in “The Model” section of the main text.

- line 80 -> bad practice of referencing to an equation that is written after.

We appreciate the comment and have fixed our equation reference.

- I think it would be appropriate to cite this interesting recent work in a similar topic, where the internal state of the system is not a phase but the radius of the particle: Togashi, Yuichi. "Modeling of Nanomachine/Micromachine Crowds: Interplay between the Internal State and Surroundings." *The Journal of Physical Chemistry B* 123.7 (2019): 1481-1490.

We thank the reviewer for pointing us towards this relevant work and have cited the paper in “The Model” section with the following text:

“The nature of our model is similar to recent work by Togashi⁴⁵, which maps an agent’s internal state / phase to its shape (radius), which affects how it interacts with its neighbors and thus the emergent collective behaviors.”

- The observables Z_G and Z_{FG} are defined from line 436 to line 446 of the Methods but they are not used in the main manuscript, maybe it would be more appropriate to define them in the SI directly, since they are only discussed there. Another option could be to include some plots of Z_G and Z_{FG} in the main manuscript, since they could be useful to describe the system's phases.

We appreciate this advice; we have decided to include order parameter plots with Z_{FG} in Supplementary Fig. 18 and define the order parameter Z_{FG} in Supplementary Discussion 7. We have kept Z_G (now referred to as Z in the revised submission) in the main text, defining it in Equation (11) and have included it in the order parameter plots in Figs. 3 and 5.

- There is also a wrong reference to the bibliography in line 333 and 335, it should be 27 instead of 28.

We thank the reviewer for pointing out this mistake and have corrected the citation numbering throughout the paper.

Final Remarks:

We appreciate all of the reviewer's concerns and have addressed the majority of issues accordingly. We believe this has resulted in a significant improvement of the clarity and quality of our work and these revisions allowed us to learn more about our system.

REVIEWER COMMENTS

Reviewer #1 (Remarks to the Author):

In the revised version of the manuscript the authors have successfully addressed all my questions.

I therefore recommend publication of the manuscript "Diverse Behaviors in Non-Uniform Chiral and Non-Chiral Swarmalators" in Nature Communications in its present form.

Reviewer #2 (Remarks to the Author):

All my comments have been addressed in this review. Overall this is a very nice paper which will be of interest to the community.

Reviewer #3 (Remarks to the Author):

-----General comment-----

I do not suggest the publication of this work in the journal "Nature Communications", since in my opinion it does not add any relevant discoveries in the field of theoretical active matter, nor in the field of modeling experimental systems. In the previous iteration of the reviewing process I asked for a more profound confrontation between experimental data and the model here presented but I cannot see any progress in this direction. As the authors themselves write in the introduction "This paper continues swarmalator research by further exploring the model introduced by O'Keefe et al", which means that no original model has been introduced, but also I cannot see any comparison with experimental evidence suggesting that the theory presented here is better than other theories in describing real biological/mechanical systems. I do not think that describing a "zoo" of simulations phenomenologies, obtained by adding an arbitrary number of parameters and new terms to a model (the original model of Ref. 27 of the manuscript), is interesting enough to be published in this journal, given its broad audience.

No progress has been made from the theoretical point of view and some misunderstandings about basic concepts of probability theory still remain (see below). I cannot see the biological relevance of this work, since the only confrontation with biological data consists in the juxtaposition of graphical representations (Fig. 8 and 9). The only attempt to compare measurable quantities is buried deep into the SI, on Fig. 59, which might hide some interesting results but it is neither explained nor commented (it also lacks units on the x-axis). No comparison with other models has been done and no correlation functions analysis of any experimental data has been done to see similarities and differences between the correlation structure in the experiments and in the model. I do not expect from a minimal model to predict exact quantities but, for the work to be interesting, I expect it at least to predict an asymptotic behavior or some features of the experiments (shape of probability distributions, exponents, shape of correlation functions etc..) that no other model can reproduce. If we stay at this superficial level of the analysis, any reasonable active matter model can qualitatively reproduce the same phenomenology here presented (Bricard, A. et al. Nat Commun 6, 7470 (2015) or James, M. et al. Nat Commun 12, 5630 (2021) or Luo-Luo Jiang et al. Phys. Rev. E 84, 021912 (2011)) hence I cannot see the usefulness of the section "Similarities to Real World Swarmalators".

From my point of view this works does not add anything to the theoretical understanding of the mechanism of syncing and swarming and neither does provide a model which is

quantitatively better in explaining an experimental evidence. I also fail to see any evident link with robotics and practical applications, but I cannot judge this point since it is not my field of study. For these reasons I do not think that this work is suitable for this journal. I think that the work of the authors, which consists of simulating a remarkable variety of possible modifications of the first swarmalator model (already theoretically studied in Ref. 27 of the manuscript) and all the improved simulations analysis is valuable, but it is suitable for publication in a more specialized journal about numerical simulations and active matter models.

-----Other issues-----

- About the quenched and dynamical noise

The authors state in the rebuttal that "However, we note that in the synchronization phenomena at least, quenched disorder and active disorder are in fact often dynamically equivalent." and then they cite two papers. I thank the authors for letting me know of this work, I was not aware of it and it is very interesting. However, in these papers they prove that Cauchy quenched noise produces a similar effect of Cauchy dynamical noise, which is a quite specific phenomenon. It may be true also in the swarmalators model as it is presented in this work but we have no solid proof of it. I can also invoke an argument of biological plausibility: if we think of the swarmalators as biological agents, it is very likely that the syncing process is not perfectly deterministic and some dynamical noise is present. For these reasons, I appreciated the inclusion of dynamical noise in the work.

- ω distributions and radius distributions

In the discussion 2 of the SI, in the third and fourth point I see the following: "there is a spread of natural frequencies across only positive values so that the whole collective has $\omega \sim U(1,3)$ $c_i=1$, and $R \sim U(1/3,1)$ ". Once again, if ω is uniformly distributed and $R=c/\omega$ (unless ω is distributed following a discrete probability distribution but this is not the case since here ω follows a uniform continuous distribution), R is not uniformly distributed. The basic change of variables shows that the distribution of R is $P(R) \sim R^{-2}$, please check the procedure to change variable in any manual of probability theory (e.g. Kolmogorov, A. N., & Bharucha-Reid, A. T. (2018). Foundations of the theory of probability: Second English Edition. Courier Dover Publications or Gardiner, Crispin W., et al. Handbook of stochastic methods. Berlin: Springer, 1985.).

I do not find Supp Fig. 2 particularly informative. If you extract your ω using the 4 prescriptions F1, F2, F3 and F4 I do not think it is necessary to plot the result of this process.

- line 139: the Greek letter ϕ is written in another fashion with respect to equation (7), please uniform the notation. I guess you also forgot to put the inverse tangent function in the definition of ϕ (it was present in the original manuscript, line 102).

- from line 444 to line 460: I think the authors wrote Fig. 7 but instead they are referring to Fig. 8

REVIEWER COMMENTS

Reviewer #1 (Remarks to the Author):

In the revised version of the manuscript the authors have successfully addressed all my questions.

I therefore recommend publication of the manuscript "Diverse Behaviors in Non-Uniform Chiral and Non-Chiral Swarmalators" in Nature Communications in its present form.

We really appreciate the reviewer's past comments that helped elevate our article's clarity and their approval for publication in Nature Communications.

Reviewer #2 (Remarks to the Author):

All my comments have been addressed in this review. Overall this is a very nice paper which will be of interest to the community.

We appreciate the reviewer's past helpful comments and their positive assessment of our work.

Reviewer #3 (Remarks to the Author):

-----General comment-----

I do not suggest the publication of this work in the journal "Nature Communications", since in my opinion it does not add any relevant discoveries in the field of theoretical active matter, nor in the field of modeling experimental systems. In the previous iteration of the reviewing process I asked for a more profound confrontation between experimental data and the model here presented but I cannot see any progress in this direction. As the authors themselves write in the introduction "This paper continues swarmalator research by further exploring the model introduced by O'Keeffe et al", which means that no original model has been introduced, but also I cannot see any comparison with experimental evidence suggesting that the theory presented here is better than other theories in describing real biological/mechanical systems. I do not think that describing a "zoo" of simulations phenomenologies, obtained by adding an arbitrary number of parameters and new terms to a model (the original model of Ref. 27 of the manuscript), is interesting enough to be published in this journal, given its broad audience.

We would like to gently push back on the reviewer's statement that the article "does not add any relevant discoveries in the theoretical active matter, nor experimental system modeling fields". First, we sincerely appreciate the reviewer's comments in previous rounds that helped significantly elevate our article's clarity and impact; in addressing those comments we learned more about our model and were able to assess the system's behavior in terms of additional order parameters, perform correlation function analysis, better describe the vortex formations, and provide a deeper understanding of the local coupling version of the model. Second, although the

latest round of comments is reasonable from a theoretical active matter point of view or modeling of experimental systems point of view, the reviewer may slightly be missing the purpose of our paper (in large part due to the language we used in the first revision of our paper, which we have since then significantly improved through both revisions). In short, the purpose of our work is to introduce and certify a swarmalator formalism that can capture universal behaviors, a framework for biologists, roboticists, engineers and other researchers to understand spatiotemporal synchronization.

We acknowledge that the previous statement “This paper continues swarmalator research by further exploring the model introduced by O’Keefe et al.” is a poor descriptor, since we introduce a new version of the model that grants several capabilities that can be switched on or off (e.g. chiral, non-chiral) and to the best of our knowledge produces more behaviors than any other active matter model. Accordingly, we have changed that sentence in the introduction so that it now reads:

This paper continues swarmalator research by introducing a new swarmalator model that facilitates diverse emergent behaviors useful to researchers in fields ranging from bio-inspired swarm robotics to microrobotics to biology; our motivation is to comprehensively map out the space of emergent behaviors the model generates.

We further appreciate the reviewer’s concern that we are adding an arbitrary number of parameters and terms to the model, but this is simply not the case. Our new swarmalator model introduces a few terms that each serve a purpose; we chose them specifically because they occur in the real world and we wanted to test the ability of our model to qualitatively reproduce real world phenomena. The model’s ability to exhibit non-chiral and chiral features means that it may now be used by researchers dealing with systems where agents’ phase behavior is an internal state oscillation (non-chiral) and systems where agents’ phase behavior is mapped to a spatial oscillatory behavior (chiral). Our frequency coupling terms are necessary because they allow our chiral swarmalators to more closely mimic some basic characteristics of counter-rotating agents. To the best of our knowledge, no other active matter model is this concise and exhibits so many different types of behaviors with global coupling. Therefore, to perform a relatively complete characterization and analysis of the new model, we studied its behavior across various natural frequency distributions (F1, F2, F3, and F4), local coupling ranges, spatial-phase coupling parameters, and levels of dynamical noise.

We have added a table (Table 1) to the Supplementary Material which gives an overview of which states are facilitated by our model and others in literature.

No progress has been made from the theoretical point of view and some misunderstandings about basic concepts of probability theory still remain (see below). I cannot see the biological relevance of this work, since the only confrontation with biological data consists in the juxtaposition of graphical representations (Fig. 8 and 9). The only attempt to compare measurable quantities is buried deep into the SI, on Fig. 59, which might hide some interesting results but it is neither explained nor commented (it also lacks units on the x-axis). No comparison with other models has been done and no correlation functions analysis of any experimental data has been done to see similarities and differences between the correlation

structure in the experiments and in the model. I do not expect from a minimal model to predict exact quantities but, for the work to be interesting, I expect it at least to predict an asymptotic behavior or some features of the experiments (shape of probability distributions, exponents, shape of correlation functions etc..) that no other model can reproduce. If we stay at this superficial level of the analysis, any reasonable active matter model can qualitatively reproduce the same phenomenology here presented (Bricard, A. et al. Nat Commun 6, 7470 (2015) or James, M. et al. Nat Commun 12, 5630 (2021) or Luo-Luo Jiang et al. Phys. Rev. E 84, 021912 (2011)) hence I cannot see the usefulness of the section "Similarities to Real World Swarmalators".

We appreciate the reviewer pointing us to the references in this comments. These references actually help our argument. They produce many of the same behaviors, but each one cannot individually be tuned to demonstrate all (or even close to all) of the behaviors which we can produce with our model. As for matches to experiments, we have added new a subfigure to show that the model reproduces the behavior of the genetic oscillators in embryonic cells. The radial phase waves (see Supplementary Fig. 59) are reproduced, and also the qualitative behavior of the order parameters; we believe this satisfies the referee's criterion that "I expect it at least to predict an asymptotic behavior or some features of the experiment". We would also like to note that the purpose of our study was to present our model; comparison with other models and the correlation structure of experimental data goes beyond the scope of our paper. The following sentence was added to the Discussion:

To the best of our knowledge, no other active matter model is this concise and offers such a diverse set of emergent collective behaviors. Indeed, many may be finely tuned to closely mimic the behavior of some real world collectives⁵¹⁻⁵³; however, with this new model we can switch between wildly different behaviors by switching just a few global parameters.

Following the reviewer's comments, we have further changed the section name "Similarities to Real World Swarmalators" to the more appropriate "Real World Parallels" to make our aim and contributions clear. We would like to add that swarm roboticists and microroboticists consistently aim for emergent behaviors enabled by minimalistic local coordination mechanisms. We have added the following line at the beginning of this section to make it clear that this section is especially useful for roboticists:

The swarmalator research field is still building momentum. While the models demonstrate rich and exciting spatiotemporal patterns³³⁻³⁸, direct use-cases continue to be elusive, earning them a 'toy-model' reputation⁴⁶. To indicate the potential of our model to break this barrier, we use this section to discuss parallels between emergent behaviors in swarmalators and a range of natural and engineered swarms. Figs. 8 and 9 show how locally coupled swarmalators qualitatively resemble emergent behaviors in slime mold, vortex arrays of spermatozoa, rotating magnetic colloids, and Quincke rollers, for example. We show additional examples in the Supplementary Material and Supplementary Movies 13-17.

From my point of view this works does not add anything to the theoretical understanding of the mechanism of syncing and swarming and neither does provide a model which is quantitatively better in explaining an experimental evidence. I also fail to see any evident link with robotics and

practical applications, but I cannot judge this point since it is not my field of study. For these reasons I do not think that this work is suitable for this journal. I think that the work of the authors, which consists of simulating a remarkable variety of possible modifications of the first swarmalator model (already theoretically studied in Ref. 27 of the manuscript) and all the improved simulations analysis is valuable, but it is suitable for publication in a more specialized journal about numerical simulations and active matter models.

Again, we sincerely appreciate all of the reviewer's detailed feedback and understand that from a purely physics perspective or an experimental system modeling perspective, our swarmalator model does not satisfy all aspects of what the researcher might be interested in seeing. Instead, it enables a diverse set of behaviors that researchers from many different fields can choose to build upon and further explore for their own use. Indeed, reviewers 1 and 2 deem this article worthy for a broad community and are content with this version of the article. In the previous revision round, reviewer 1 stated that "this study lays a solid foundation for future research activities in areas as diverse as active matter, agents self-organisation, micro-robotics, etc." and reviewer 2 stated that "it will provide a useful tool for swarm engineers, biologists, and micro-engineers alike." This is exactly what we aim for with our swarmalator model and article, to demonstrate diverse behaviors, characterize the vast majority, and qualitatively compare some to real-world systems so that researchers in micro-robotics, bio-inspired swarm robotics, self-assembly, biology, and many other fields can use our work as a stepping stone to guide design of new systems or characterization of existing ones.

We further believe that roboticists, especially those operating at micro-scales, will appreciate our work, because it demonstrates that a diverse set of dynamic and stationary collective behaviors can result from a minimalistic coupling model. This is very powerful because it means that agents can rely on limited state information and sensing, yet produce a range of emergent, stable behaviors. Future microrobotic swarms are expected to revolutionize biomedicine (drug delivery and tetherless surgery) and manufacturing (self-assembly and programmable materials). Emergent behaviors in minimalistic robot swarms is currently explored at all length scales by many researchers ranging from engineering to biology to physics to computer science. Platforms based on robot swarms are also currently being used by entomologists and physicists to study complex emergent behaviors in living and engineered systems. We reference here a couple of papers along that line of research: Li, S., *et al.* Particle robotics based on statistical mechanics of loosely coupled components. *Nature* **567**, 361–365 (2019); Savoie, W. et al. A robot made of robots: Emergent transport and control of a smarticle ensemble. *Science Robotics*, Vol. 4, Issue 34 (2019); Zhou W. et al. Collective Synchronization of Undulatory Movement through Contact. *Physical Review X* 11, 031051. We have edited the Discussion section to highlight our model's relevance to robotics:

Given the universality of phase dependence and circular motion, this model may be used to advance studies across fields, from inspiring new theoretical advances in active matter to modeling frameworks in developmental and behavioral biology. These minimalistic coordination schemes may also support macro-scale robot swarms where constituents travel in periodic orbits that exceed their range of communication^{43,51}, multi-robot systems acting in the human space where more advanced perception and reasoning presents security concerns⁵², and microrobot swarms in biomedical applications which are fundamentally limited in both sensing and

computation power⁵³⁻⁵⁵. We hope this encourages a line of fundamental studies on the emergent behaviors that result from an interdependence between time-domain and spatial-domain-specific parameters.

We would also like to appreciate and highlight the reviewer's statement that our simulation analysis is valuable; however, we believe that the work should be highlighted in an interdisciplinary journal like Nature Communications since we present a new swarmalator model and find so many new behaviors of interest to so many different fields through that single model.

We would like to make a final note, again, that we appreciate all of the critique from the reviewer. Especially during the first revision round, this was necessary for a complete characterization and analysis of our new model. In the previous revision, we added to our manuscript a substantial amount of material that characterized the new swarmalator model's behavior. Below we have listed the material we added from the previous revision which makes our model useful for diverse research communities.

- Incorporation of dynamical noise
- Correlation function analysis
- Vortex characterization
- Local coupling characterization
- Comparison with experimental data
- Order parameter plots to distinguish between different states in global coupling

As shown through this list, we have added a significant amount of material that clearly enhances the characterization and analysis of our new model so that it is useful for researchers in diverse fields. We believe that more comparison to real world collectives and/or theoretical analysis goes beyond the scope of this paper.

-----Other issues-----

- About the quenched and dynamical noise

The authors state in the rebuttal that "However, we note that in the synchronization phenomena at least, quenched disorder and active disorder are in fact often dynamically equivalent." and then they cite two papers. I thank the authors for letting me know of this work, I was not aware of it and it is very interesting. However, in these papers they prove that Cauchy quenched noise produces a similar effect of Cauchy dynamical noise, which is a quite specific phenomenon. It

may be true also in the swarmalators model as it is presented in this work but we have no solid proof of it. I can also invoke an argument of biological plausibility: if we think of the swarmalators as biological agents, it is very likely that the syncing process is not perfectly deterministic and some dynamical noise is present. For these reasons, I appreciated the inclusion of dynamical noise in the work.

We appreciate the reviewer's previous suggestion that we include dynamical noise in our work and agree that our inclusion of it during the previous revision round could make it useful for studying biological collectives.

- ω distributions and radius distributions

In the discussion 2 of the SI, in the third and fourth point I see the following: "there is a spread of natural frequencies across only positive values so that the whole collective has $\omega \sim U(1,3)$ $c_i=1$, and $R \sim U(1/3,1)$ ". Once again, if ω is uniformly distributed and $R=c/\omega$ (unless ω is distributed following a discrete probability distribution but this is not the case since here ω follows a uniform continuous distribution), R is not uniformly distributed. The basic change of variables shows that the distribution of R is $P(R) \sim R^{-2}$, please check the procedure to change variable in any manual of probability theory (e.g. Kolmogorov, A. N., & Bharucha-Reid, A. T. (2018). Foundations of the theory of probability: Second English Edition. Courier Dover Publications or Gardiner, Crispin W., et al. Handbook of stochastic methods. Berlin: Springer, 1985.).

I do not find Supp Fig. 2 particularly informative. If you extract your ω using the 4 prescriptions F1, F2, F3 and F4 I do not think it is necessary to plot the result of this process.

We thank the reviewer for pointing out this error in our notation of the distribution of R , and have corrected the notation. During the previous revision round, we changed the way in which we generate the natural frequencies so that the distributions of the ω would be more realistic to biological or robot collectives' behavior. We have modified Supplementary Fig. 2 (shown in the following figure) so that the reader can clearly see the distributions of ω and R . Since this study is intended for researchers in a wide variety of fields, we suspect that the distribution behavior of ω and R may not be immediately intuitive, and thus it will increase the paper's clarity.

Supplementary Fig. 2. Natural frequency distributions. Histogram for each of the natural frequency distributions used throughout the numerical studies. (a-b) $F1$; (c-d) $F2$; (e-f) $F3$; (g-h) $F4$. The distributions shown in (a,c,e,g) correspond to the natural frequencies and the distributions shown in (b,d,f,h) correspond to radius of revolution.

- line 139: the Greek letter ϕ is written in another fashion with respect to equation (7), please uniform the notation. I guess you also forgot to put the inverse tangent function in the definition of ϕ (it was present in the original manuscript, line 102).

We have corrected this mistake in the manuscript.

- from line 444 to line 460: I think the authors wrote Fig. 7 but instead they are referring to Fig. 8

We appreciate the reviewer's strong attention to detail throughout the review process; we have fixed this mistake and have read through the manuscript several times to make sure the correct figures and equations are referenced.

REVIEWERS' COMMENTS

Reviewer #1 (Remarks to the Author):

The revised version of manuscript "Diverse Behaviours in Non-Uniform Chiral and Non-Chiral Swarmalators", by S. Ceron et al. satisfactory addressed all the concerns of Reviewer #3. Based on this, I recommend publication of this work in Nature Communications in its present form. Below I provide my arguments in support of this decision.

Reviewer #3 raised three main concerns:

1. "I do not suggest the publication of this work in the journal "Nature Communications", since in my opinion it does not add any relevant discoveries in the field of theoretical active matter, nor in the field of modelling experimental systems."

This is a typical example of a subjective statement, which expresses the personal taste of the Reviewer, rather than an objective discussion of the reported results, which indeed include several interesting novel steady states. The main value and potential impact of this study is in the systematic investigation of the "phase" (steady state) behaviour of the proposed model, as is shown in Figs. 2 and 4. Usually, these types of results, summarizing universal qualitative, rather than quantitative aspects of a system under investigation, are of interest for a general readership. In contrast, non-universal characteristics, such as correlations functions, would be of interest, for more specialized readers.

2. Next, Referee #3 writes: "If we stay at this superficial level of the analysis, any reasonable active matter model can qualitatively reproduce the same phenomenology here presented (Bricard, A. et al. Nat Commun 6, 7470 (2015) or James, M. et al. Nat Commun 12, 5630 (2021) or Luo-Luo Jiang et al. Phys. Rev. E 84, 021912 (2011))."

This concern is a mixture of a subjective opinion and an empty statement. Indeed, all the three referred articles deal with completely different systems and the models proposed were constructed to address specific experimental, or theoretical questions. It is not clear that these active matter models can also exhibit such a rich collective behaviour as reported in the present manuscript by S. Ceron et al. The claim of Referee #3 that the earlier models "can qualitatively reproduce the same phenomenology here presented" is not justified by any actual calculations, and therefore it is just an empty phrase. In contrast, the authors of the manuscript under review, proposed a physically motivated model of active matter, with a clear aim to investigate the physical consequences of the model assumptions on emergent collective behaviour. The model has been thoroughly analysed and revealed new and rich phenomenology.

3. Finally, Referee #3 concluded: "I think that the work of the authors, which consists of simulating a remarkable variety of possible modifications of the first swarmalator model (already theoretically studied in Ref. 27 of the manuscript) and all the improved simulations analysis is valuable, but it is suitable for publication in a more specialized journal about numerical simulations and active matter models."

This concern of Referee #3 is also not justified, indeed the present model is a significant, physically motivated and non-trivial extension of the earlier research in [27]. The model generalizations include: i) non-identical and chiral agents, ii) short-ranged interaction in the real space, which enabled a range of novel non-trivial time-dependent configurations, e.g., bouncing and rotating clusters, and various types of self-organized phase configurations.

Reviewer #2 (Remarks to the Author):

Thank you for this additional round of reviews. From the start I have liked this paper and believe it to be of interest to a broad readership. Since then substantial modifications have been made that further strengthen the paper and address sensitivities of different research communities.

Reviewer #3 (Remarks to the Author):

I appreciate the answers from the authors and I really think that they made an excellent job in characterizing all the different behaviors of the swarmalators model.

However, I still think that this research has not the sufficient impact to be published in this journal. This is my opinion. Given that, I completely put the final decision in the hands of the editor.

REVIEWERS' COMMENTS

Reviewer #1 (Remarks to the Author):

The revised version of manuscript "Diverse Behaviours in Non-Uniform Chiral and Non-Chiral Swarmalators", by S. Ceron et al. satisfactory addressed all the concerns of Reviewer #3. Based on this, I recommend publication of this work in Nature Communications in its present form. Below I provide my arguments in support of this decision.

We appreciate the reviewer's recommendation of our work and their argument in support of our work. We would also like to thank the reviewer's comments in previous revisions which helped elevate the clarity and impact of our work.

Reviewer #3 raised three main concerns:

1. "I do not suggest the publication of this work in the journal "Nature Communications", since in my opinion it does not add any relevant discoveries in the field of theoretical active matter, nor in the field of modelling experimental systems."

This is a typical example of a subjective statement, which expresses the personal taste of the Reviewer, rather than an objective discussion of the reported results, which indeed include several interesting novel steady states. The main value and potential impact of this study is in the systematic investigation of the "phase" (steady state) behaviour of the proposed model, as is shown in Figs. 2 and 4. Usually, these types of results, summarizing universal qualitative, rather than quantitative aspects of a system under investigation, are of interest for a general readership. In contrast, non-universal characteristics, such as correlations functions, would be of interest, for more specialized readers.

2. Next, Referee #3 writes: "If we stay at this superficial level of the analysis, any reasonable active matter model can qualitatively reproduce the same phenomenology here presented (Bricard, A. et al. Nat Commun 6, 7470 (2015) or James, M. et al. Nat Commun 12, 5630 (2021) or Luo-Luo Jiang et al. Phys. Rev. E 84, 021912 (2011))."

This concern is a mixture of a subjective opinion and an empty statement. Indeed, all the three referred articles deal with completely different systems and the models proposed were constructed to address specific experimental, or theoretical questions. It is not clear that these active matter models can also exhibit such a rich collective behaviour as reported in the present manuscript by S. Ceron et al. The claim of Referee #3 that the earlier models "can qualitatively reproduce the same phenomenology here presented" is not justified by any actual calculations, and therefore it is just an empty phrase. In contrast, the authors of the manuscript under review, proposed a physically motivated model of active matter, with a clear aim to investigate the physical consequences of the model assumptions on emergent collective behaviour. The model has been thoroughly analysed and revealed new and rich phenomenology.

3. Finally, Referee #3 concluded: "I think that the work of the authors, which consists of simulating a remarkable variety of possible modifications of the first swarmalator model (already theoretically studied in Ref. 27 of the manuscript) and all the improved simulations analysis is valuable, but it is suitable for

publication in a more specialized journal about numerical simulations and active matter models."

This concern of Referee #3 is also not justified, indeed the present model is a significant, physically motivated and non-trivial extension of the earlier research in [27]. The model generalizations include: i) non-identical and chiral agents, ii) short-ranged interaction in the real space, which enabled a range of novel non-trivial time-dependent configurations, e.g., bouncing and rotating clusters, and various types of self-organized phase configurations.

Reviewer #2 (Remarks to the Author):

Thank you for this additional round of reviews. From the start I have liked this paper and believe it to be of interest to a broad readership. Since then substantial modifications have been made that further strengthen the paper and address sensitivities of different research communities.

We appreciate the reviewer's positive assessment of our work and agree that our modifications will be useful to many readers in diverse research areas.

Reviewer #3 (Remarks to the Author):

I appreciate the answers from the authors and I really think that they made an excellent job in characterizing all the different behaviors of the swarmalators model.

However, I still think that this research has not the sufficient impact to be published in this journal. This is my opinion. Given that, I completely put the final decision in the hands of the editor.

We appreciate the reviewer's comments throughout this process. As we have stated in the past, the reviewer's thoroughness was critical in elevating the clarity and impact of our article.